

# Efficient production of singlet oxygen and organic triplet excited states in aqueous PM$_{2.5}$ in Hong Kong, South China

Yuting Lyu[1], Yin Hau Lam[1], Yitao Li[1], Nadine Borduas-Dedekind[2], and Theodora Nah[1,3]

[1]School of Energy and Environment, City University of Hong Kong, Hong Kong SAR, China
[2]Department of Chemistry, University of British Columbia, Vancouver V6T 1Z1, Canada
[3]State Key Laboratory of Marine Pollution, City University of Hong Kong, Hong Kong SAR, China

**Correspondence:** Theodora Nah (theodora.nah@cityu.edu.hk)

**Abstract.** Photooxidants drive many atmospheric chemical processes. The photoexcitation of light-absorbing organic compounds (i.e., brown carbon (BrC)) in atmospheric waters can lead to the generation of reactive organic triplet excited states ($^3C^*$), which can undergo further reactions to produce other photooxidants such as singlet oxygen ($^1O_2$). To determine the importance of these aqueous photooxidants in SOA formation and transformation, we must know their steady-state concen-
trations and quantum yields. However, there has been limited measurements of aqueous $^3C^*$ and $^1O_2$ in atmospheric samples outside of North America and Europe. In this work, we report the first measurements of the steady-state concentrations and quantum yields of $^3C^*$ and $^1O_2$ produced in aerosols in South China. We quantified the production of $^3C^*$ and $^1O_2$ in illuminated aqueous extracts of PM$_{2.5}$ collected in different seasons at two urban sites and one coastal semi-rural site during a year-round study conducted in Hong Kong, South China. The mass absorption coefficients at 300 nm for BrC in the aqueous
PM$_{2.5}$ extracts ranged from $0.49 \times 10^4$ to $2.01 \times 10^4$ cm$^2$ g-C$^{-1}$ for the three sites. Both $^1O_2$ and $^3C^*$ were produced year-round. The steady-state concentrations of $^1O_2$ ($[^1O_2]_{ss}$) in the illuminated aqueous extracts spanned two orders of magnitude, ranging from $1.56 \times 10^{-14}$ to $1.35 \times 10^{-12}$ M, with a study average of $(4.02 \pm 3.52) \times 10^{-13}$ M. The steady-state concentrations of $^3C^*$ ($[^3C^*]_{ss}$) in the illuminated aqueous extracts spanned two orders of magnitude, ranging from $2.93 \times 10^{-16}$ to $8.08 \times 10^{-14}$ M, with a study average of $(1.09 \pm 1.39) \times 10^{-14}$ M. The $[^1O_2]_{ss}$ and $[^3C^*]_{ss}$ correlated with the concentration and
absorbance of BrC, thus implying that the amount of BrC drives the steady-state concentrations of these photooxidants. The locations (urban vs. semi-rural) did not have a significant effect on $[^3C^*]_{ss}$ and $[^1O_2]_{ss}$, which indicated that BrC from local sources did not have a significant influence on the year-round $^3C^*$ and $^1O_2$ production. $^3C^*$ and $^1O_2$ production were found to be the highest in winter and the lowest in summer for all three sites. The observed seasonal trends of $^1O_2$ and $^3C^*$ production could be attributed to the seasonal variations in long-range air mass transport. Our analysis highlighted the key role that re-
gional sources play in influencing the composition and concentrations of water-soluble BrC in winter PM$_{2.5}$ in Hong Kong, which contributed to their highest $^3C^*$ and $^1O_2$ production. The current results will be useful for modeling seasonal aqueous organic aerosol photochemistry in the South China region.





## 1 Introduction

Atmospheric aqueous phases (e.g., aqueous aerosol, cloud water, fog droplets) serve as important media for chemical reactions
of organic compounds. Many of the chemical transformations in atmospheric aqueous phases are driven by photochemically
generated oxidants, particularly triplet excited states of organic matter ($^3C^*$), singlet state oxygen ($^1O_2$), and hydroxyl radicals
($\cdot OH$). Light absorbing organic compounds, commonly known as brown carbon (BrC), serve as key precursors for the formation
of photooxidants in atmospheric aqueous phases (Laskin et al., 2015; Hems et al., 2021).

Upon the absorption of sunlight, some BrC chromophores (e.g., aromatic carbonyls) can be promoted from their ground
states to reactive $^3C^*$ with species-specific energy levels (Canonica et al., 1995; Yu et al., 2014). $^3C^*$ is not a single photoox-
idant. Instead, $^3C^*$ is comprised of a variety of species with a range of reactivities (McNeill and Canonica, 2016). Some $^3C^*$
species can react rapidly with organic compounds (e.g., phenolic compounds and anilines) through single-electron transfer
and proton-coupled electron transfer reactions (Lathioor and Leigh, 2006; Erickson et al., 2015). Some $^3C^*$ species can also
react with organic compounds (e.g., aromatic amino acids) through hydrogen abstraction reactions (Walling and Gibian, 1965;
Tsentalovich et al., 2002). In addition, energy transfer from $^3C^*$ to molecular oxygen ($^3O_2$) leads to the formation of $^1O_2$
(Herzberg and Herzberg, 1947). This reaction occurs rapidly under ambient conditions for most $^3C^*$ species since the energy
required for $^3O_2 \rightarrow {}^1\Delta_g$ is only 94 kJ mol$^{-1}$ (Zepp et al., 1985; Wilkinson et al., 1993; McNeill and Canonica, 2016). $^1O_2$
typically reacts with electron-rich/unsaturated species (e.g., alkenes, cyclic dienes, polycyclic aromatic hydrocarbons) through
addition reactions (Ghogare and Greer, 2016; Kaur and Anastasio, 2017; Nolte and Peijnenburg, 2018; Manfrin et al., 2019;
Barrios et al., 2021). The production of $^3C^*$ and $^1O_2$ are influenced by both the concentrations (i.e., quantity) and specific
absorbance (i.e., quality) of BrC chromophores (Bogler et al., 2022). The relative importance of the quantity vs. quality of BrC
chromophores in the production of $^3C^*$ and $^1O_2$ depends on the BrC source.

Aqueous reactions between organic compounds and photooxidants play key roles in forming and transforming secondary
organic aerosols (SOA). Understanding the significance and contributions of these reactions to the SOA budget necessitates
knowledge of the steady-state concentrations and quantum yields of the photooxidants. Out of all the photooxidants, $\cdot OH$
production in various atmospheric aqueous phases has been the most widely investigated (Arakaki and Faust, 1998; Arakaki
et al., 1999, 2006, 2013; Anastasio and McGregor, 2001; Anastasio and Jordan, 2004; Anastasio and Newberg, 2007; Kaur and
Anastasio, 2017; Kaur et al., 2019; Manfrin et al., 2019; Leresche et al., 2021; Ma et al., 2023a). $\cdot OH$ can be photochemically
produced from BrC (Chen et al., 2021; Li et al., 2022) and other photolabile compounds such as inorganic nitrate, nitrite, and
metal-organic complexes (Kaur and Anastasio, 2017; Kaur et al., 2019; Leresche et al., 2021; Ma et al., 2023a). There has been
considerably fewer measurements of $^3C^*$ and $^1O_2$ production in atmospheric aqueous phases.

So far, several studies have measured $^1O_2$ production in cloud water (Faust and Allen, 1992), fog water (Anastasio and
McGregor, 2001; Kaur and Anastasio, 2017), rain water (Albinet et al., 2010), and particulate matter (PM) extracts (Cote et al.,
2018; Kaur et al., 2019; Manfrin et al., 2019; Leresche et al., 2021; Bogler et al., 2022; Ma et al., 2023a). $^1O_2$ originates
from a $^3C^*$ molecule, and therefore measuring both $^1O_2$ and its $^3C^*$ precursor is important. However, there has only been four
investigations of $^3C^*$ production in atmospheric aqueous phases (Kaur and Anastasio, 2018; Kaur et al., 2019; Chen et al.,



2021; Ma et al., 2023a). These studies showed that the concentrations of $^3C^*$ ($10^{-16}$ to $10^{-13}$ M) and $^1O_2$ ($10^{-14}$ to $10^{-12}$ M) produced are typically 2 to 4 orders of magnitude larger than the concentrations of ·OH ($10^{-17}$ to $10^{-15}$ M) produced. Thus, despite the reactivity of $^3C^*$ and $^1O_2$ being substantially lower than ·OH, $^3C^*$ and $^1O_2$ can play important roles in aqueous
SOA formation and transformation due to their large concentrations.

Spatiotemporal measurements of photooxidant production in atmospheric aqueous phases are important to understand how aqueous reactions between organic compounds and photooxidants can change as a function of season and of location. Leresche et al. (2021) measured ·OH and $^1O_2$ production in illuminated extracts of $PM_{2.5}$ collected during the winter, spring, and summer seasons in urban and rural settings in Colorado, USA, while Bogler et al. (2022) measured $^1O_2$ production in illuminated
extracts of $PM_{10}$ collected year-round at a rural site and a suburban site in Switzerland. The two studies highlighted the roles that seasonality and/or local anthropogenic activities play in influencing photooxidant production. At present, investigations of photooxidant production in atmospheric aqueous phases have been restricted to North America and Europe. Given the important role that aqueous photochemistry plays in forming and transforming SOA in many regions, there is, therefore, a need to investigate the spatiotemporal variations of photooxidant production in atmospheric aqueous phases in regions outside of
North America and Europe.

In this work, we investigated the production of $^3C^*$ and $^1O_2$ in illuminated extracts of $PM_{2.5}$ collected during different seasons at three sites (two urban and one semi-rural) in Hong Kong. Hong Kong is a densely populated coastal city located on the east of the Pearl River Delta (PRD) in South China. Its seasonal meteorological conditions and air quality are strongly influenced by the East Asian monsoon (Yihui and Chan, 2005). Clean, marine air masses are transported from southwestern sea
areas to Hong Kong in the summer, whereas polluted air mass are transported from northern continental areas to Hong Kong in mid-fall and winter (Tanner and Law, 2002). Local sources are the main contributors to summer $PM_{2.5}$, whereas regional sources are the main contributors to winter $PM_{2.5}$ (Pathak et al., 2003; Louie et al., 2005; Louie, 2005; Huang et al., 2014; Li et al., 2015; Wong et al., 2020). The main objectives of this study are to (1) characterize the steady-state concentrations and apparent quantum yields of $^3C^*$ and $^1O_2$, and (2) determine how location and seasonality influence $^3C^*$ and $^1O_2$ production in
Hong Kong. This work presents the first spatiotemporal measurements of photooxidants produced in atmospheric aerosols in East Asia. Results from this study provide insights into the levels of $^3C^*$ and $^1O_2$ produced in $PM_{2.5}$ in the South China region, which will be useful for improving our understanding of aqueous organic aerosol photochemical processes in this region.

## 2 Methods

### 2.1 $PM_{2.5}$ filter sampling and extraction

#### 2.1.1 Sampling locations

The year-round sampling campaign took place from December 2020 to December 2021 in Hong Kong. The three sites were the City University of Hong Kong campus (CU, 22°20'05"N, 114°10'23"E), and the air quality monitoring stations at Tsuen Wan (TW, 22°20'17"N, 114°06'52"E) and Hok Tsui (HT, 22°12'33"N, 114°15'12"E) (Figure 1). The CU and TW sites are located



in urban areas with many residential and commercial (and industrial for TW) activities. Since the semi-rural coastal HT site

is located away from local emission sources (approximately 6 km away from the closest urban area), it was mostly used as a receptor site to monitor air pollution originating from sources outside of Hong Kong in past studies (Tanner and Law, 2002; Li et al., 2018). In Hong Kong, winter nominally runs from December to February, spring runs from March to May, summer runs from June to August, and fall runs from September to November. Sampling activities at each site took place for approximately one month during each season (Table S1).

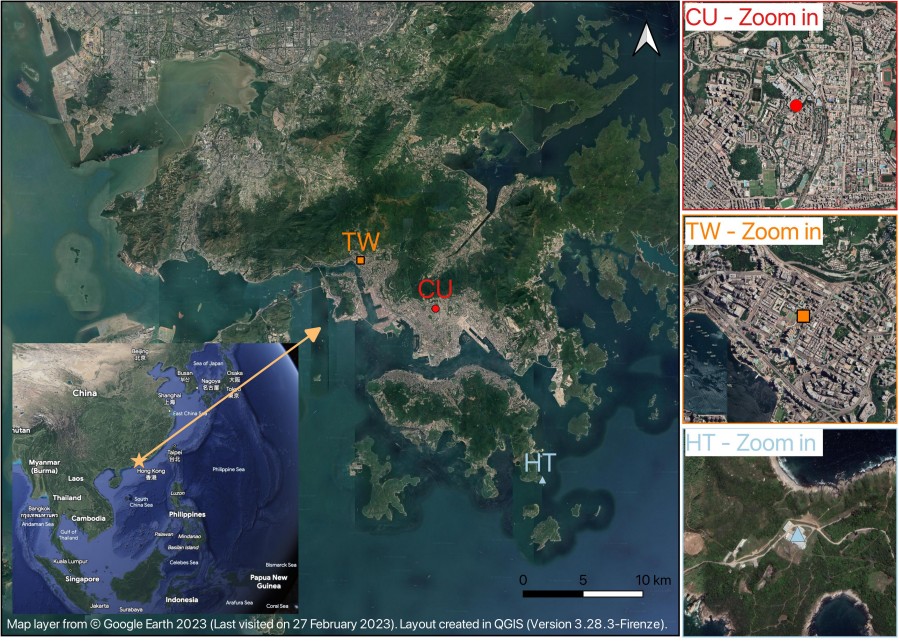

**Figure 1.** Satellite image of the three sites in Hong Kong. CU, TW, HT are short for the City University of Hong Kong campus, Tsuen Wan, and Hok Tsui sites, respectively.

**2.1.2   Sampling and extraction protocols**

$PM_{2.5}$ was collected on three prebaked (550 °C for 12 h) 47 mm diameter quartz filters (Pall Tissuquartz$^{TM}$, 2500 QAT-UP) using a custom-built medium-volume sampler with a $PM_{2.5}$ inlet. Ambient air was sampled onto each quartz filter at 30 L min$^{-1}$. The sampler was deployed at ground level at the CU and HT sites, and on a 17 m rooftop at the TW site. $PM_{2.5}$ samples were collected continuously for 72 hours every third day. The filter samples were stored in resealable bags at -25 °C

until further processing. Blank filter samples were generated the same way as the ambient filter samples, except the sampler pump for this channel was switched off during sampling.

Each filter was extracted in Milli-Q water inside a sterile centrifuge tube (JET BIOFIL$^{®}$) by vortexing (MX-S DLAB, medium high power). The disintegrated filter parts were removed from the extracts by filtration using 0.22 µm pore size nylon



syringe filters (Nylon66, Jinteng®). The filtered extracts were stored in amber vials at 4 °C in a refrigerator until use. To ensure
that there was sufficient PM$_{2.5}$ material for photochemical experiments and chemical analysis, extracts from three consecutive
sampling periods (9 days in total) were aggregated and then diluted to an adequate volume. This procedure resulted in roughly
3 aggregated extracts per season for each site, referred to by the site and sampling start date. For example, CU041220 refers to
extracts of filters collected from 4 Dec 2020 to 13 Dec 2020 at the CU site. Detailed information about the sampling periods
and allocation of aggregated extracts are listed in Table S1. Due to sampler pump malfunction, filters were not collected at
the CU site from 18 June 2020 to 24 June 2020 and at the HT site from 18 April 2020 to 27 April 2020. In addition, some
aggregated extracts were comprised only of two consecutive sampling periods (6 days in total) due to limited filter samples.

## 2.2  Light absorption measurements

The UV-Visible absorbance spectra of the extracts were obtained in 1-nm increments using a UV-VIS-NIR spectrophotometer
(Shimadzu UV-3600) with Milli-Q water as the reference sample. The spectra were corrected by subtracting spectra from the
field blanks and the average absorbance between 700 to 800 nm (Ossola et al., 2021). The decadic absorption coefficient ($\alpha_\lambda$,
cm$^{-1}$) was calculated using the following equation:

$$\alpha_\lambda = \frac{A_\lambda}{l} \tag{1}$$

where $A_\lambda$ is the dimensionless absorbance of extracts at wavelength $\lambda$, and $l$ is the optical path length (1 cm) of the cuvette.
The rate of light absorption (R$_{abs}$, mol-photons L$^{-1}$ s$^{-1}$) of each extract was calculated using the following equation:

$$R_{abs} = \frac{10^3}{d} \sum_{290\,nm}^{600\,nm} I_{0,\lambda}(1 - 10^{-\alpha_\lambda d})\Delta\lambda \tag{2}$$

where $d$ is the optical path length of the quartz tubes used in the photochemical experiments (1.25 cm), $10^3$ is for units
conversion (cm$^3$ L$^{-1}$), $I_{0,\lambda}$ (mol-photons nm$^{-1}$ s$^{-1}$ cm$^{-2}$) is the absolute irradiance of the light source at wavelength $\lambda$, and
$\Delta\lambda$ is the interval of wavelength (1 nm). A wavelength range of 290 to 600 nm was used to cover both the bandwidth and light
absorption range of all the extracts (Figure S1). R$_{abs}$ was not corrected for light screening (i.e., inner filter effect) since the
absorbance coefficients of all the extracts were below 0.1 cm$^{-1}$ in the UVA range. The wavelength-dependent mass absorption
coefficients for the WSOC (MAC$_\lambda$) in the extracts were calculated using the following equation:

$$MAC_\lambda = \frac{\alpha_\lambda \times \ln(10)}{[WSOC] \times 10^{-6}} \tag{3}$$

where $\ln(10)$ is the base conversion factor, $10^6$ is for unit conversion (mg L$^{-1}$ to g cm$^{-3}$), and [WSOC] (in mg-C L$^{-1}$) is
the concentration of the WSOC in each extract (Table S2) measured by a TOC Analyzer (Shimadzu TOC-V CSH). It should
be noted that the mass ratio of the organic material (OM) to organic carbon (OC) in PM$_{2.5}$ in Hong Kong is approximately
2.1 (Chen and Yu, 2007). Thus, the calculated MAC$_\lambda$ values would be halved had they been normalized by [OM] instead of
[WSOC]. Section S1 describes the other chemical analysis performed on the extracts.

Various optical parameters were obtained for each extract based on their absorbance and WSOC measurements (Table S3).
The $\alpha_{300}$ is the UV absorption coefficients at 300 nm. SUVA$_{254}$ and SUVA$_{365}$ are the specific UV absorbance obtained from



dividing the UV absorption coefficients at 254 nm and at 365 nm ($\alpha_{254}$ and $\alpha_{365}$, respectively) by [WSOC]. The AAE is the absorption Ångström exponent, which can be calculated using the following equation:

$$\text{AAE} = -\frac{\ln(\alpha_{\lambda_2}/\alpha_{\lambda_1})}{\ln(\lambda_2/\lambda_1)} \tag{4}$$

The AAE values were obtained from the negative of the slope of the linear plot of $\ln(\alpha_\lambda)$ vs. $\ln(\lambda)$ in the range of 300 to 450 nm (26 extracts) or 300 to 350 nm (8 extracts). The narrower wavelength range was used for extracts that had very low absorbance at the long wavelengths to ensure good linearity.

### 2.3 Chemicals used in photochemical experiments

The chemical probe for $^1O_2$ , furfuryl alcohol (98 %), was purchased from Acros Organics and was distilled under vacuum before being prepared into a 100 µM stock solution. Deuterium oxide ($D_2O$, 99 % atom D) was purchased from Sigma Aldrich. The chemical probe for $^3C^*$ , 2,6-dimethoxyphenol (syringol, 98 %), was purchased from J&K Scientific. The chemical actinometer, 2-nitrobenzaldhyde (2-NB, 98 %), was purchased from J&K Scientific. Preparation of all chemical solutions and dilution of the extracts were performed using ultrapure water (Milli-Q, Merck, resistivity of 18.2 MΩ cm).

### 2.4 Photochemical experiments

Irradiation experiments were conducted in a Rayonet photoreactor (RPR-200, Southern New England Ultraviolet Co.) equipped with 12 UVA lamps (RPR-3500Å, Southern New England Ultraviolet Co.). The spectral irradiance is shown in Figure S2. The procedure used to determine the photon flux is described in Section S2. In a typical photochemical experiment, quartz tubes containing 5 mL of extract spiked with a probe compound (10 µM) were placed on a merry-go-round sample holder (RMA-500, Southern New England Ultraviolet Co.) in the middle of the photoreactor for continuous illumination. The chemical probes for $^1O_2$ and $^3C^*$ were furfuryl alcohol (Appiani et al., 2017) and syringol (Kaur and Anastasio, 2018; Kaur et al., 2019; Ma et al., 2023a), respectively. The temperature inside the photoreactor during the experiment was maintained at 26 ± 1 °C by a cooling fan positioned at the bottom of the photoreactor. Aliquots of the solutions were removed at different reaction times to monitor the loss of the chemical probe using a ultrahigh-pressure liquid chromatography system coupled to a photodiode array detector (UPLC-PDA, Waters ACQUITY H-Class). Separation of the chemical probes, furfuryl alcohol and syringol, was achieved using a Phenomenex Kinetex polar C18 column (2.6 µm, 100 × 2.1 mm) and elution at 0.3 mL min$^{-1}$ with Milli-Q water/acetonitrile ratios of 9:1 and 8:2, respectively. The PDA detection wavelengths for furfuryl alcohol and syringol were 216 nm and 210 nm, respectively. Control experiments showed that syringol and furfuryl alcohol loss in illuminated Milli-Q water and field blank extracts were mostly minimal and the differences were within experimental errors (Figure S3). This indicated that $^3C^*$ and $^1O_2$ production were negligible in the background samples.

### 2.5 Quantification of steady-state concentrations, formation rates, and quantum yields of $^1O_2$

Furfuryl alcohol was used as the $^1O_2$ chemical probe (Appiani et al., 2017). The kinetic solvent isotope effect (KSIE) was used to account for furfuryl alcohol degradation by oxidants other than $^1O_2$ in the quantification of the steady-state concentrations



of $^1O_2$ ($[^1O_2]_{ss}$) in the extracts (Davis et al., 2018). These experiments involved comparing the decays of furfuryl alcohol in pure water ($H_2O$) vs. in heavy water ($D_2O$) (Haag and Hoigne, 1986; Allen et al., 1996; Anastasio and McGregor, 2001; Kaur and Anastasio, 2017; Kaur et al., 2019; Ma et al., 2023a). The extracts were prepared in Milli-Q water or in a mixture of 1:1 Milli-Q water/$D_2O$ (v/v), wherein they were spiked with 10 μM furfuryl alcohol. The furfuryl alcohol decay followed pseudo

first order kinetics (Figure S4). Their rate constants were used to calculate $[^1O_2]_{ss}$ as follows:

$$[^1O_2]_{ss} = \frac{k'_{FFA,D_2O} - k'_{FFA,H_2O}}{k^{FFA+^1O_2}_{rxn} \times \frac{k_{d,H_2O} - k_{d,D_2O}}{k_{d,H_2O} + k_{d,D_2O}}} \qquad (5)$$

where $k'_{FFA,D_2O}$ and $k'_{FFA,H_2O}$ are the pseudo first order rate constants of furfuryl alcohol loss in the 1:1 Milli-Q water/$D_2O$ (v/v) mixture and in Milli-Q water, respectively, determined from the slopes of the linear plot of $\ln([FFA]_t/[FFA]_0)$ vs. irradiation time (Figure S4), $k^{FFA+^1O_2}_{rxn}$ is the second order rate constant of FFA with $^1O_2$ at 26 °C ($1.084 \times 10^8$ $M^{-1}$ $s^{-1}$) (Appiani et al.,

2017), and $k_{d,H_2O}$ and $k_{d,D_2O}$ are the $^1O_2$ deactivation rates in pure $H_2O$ ($2.81 \times 10^5$ $s^{-1}$) and pure $D_2O$ ($1.57 \times 10^4$ $s^{-1}$), respectively (Davis et al., 2018). Since the furfuryl alcohol decay from direct photolysis was minimal, the photolysis rate ($7.58 \pm 0.83 \times 10^{-7}$ $s^{-1}$, Figure S3) was not used to correct the $k'_{obs,D_2O}$ and $k'_{obs,H_2O}$ values.

The formation rate of $^1O_2$ ($R_{f,^1O_2}$) was calculated as follows:

$$R_{f,^1O_2} = [^1O_2]_{ss} \times k_{d,H_2O} \qquad (6)$$

The apparent quantum yield of $^1O_2$ ($\Phi_{^1O_2}$) was calculated as follows:

$$\Phi_{^1O_2} = \frac{R_{f,^1O_2}}{R_{abs}} \qquad (7)$$

## 2.6   Quantification of steady-state concentrations, formation rates, and quantum yields of $^3C^*$

We used syringol as the sole $^3C^*$ chemical probe. However, we acknowledge that due to the chemical complexity of $^3C^*$ species, a single chemical probe has limitations to quantify all the $^3C^*$ species (McNeill and Canonica, 2016; Maizel and

Remucal, 2017). While some studies have used multiple probes (and thus, performed multiple photochemical experiments) to better constrain $^3C^*$ measurements (Kaur and Anastasio, 2018; Kaur et al., 2019; Ma et al., 2023a), we were unable to do so in our study due to insufficient extract volumes for additional photochemical experiments. Thus, only a subset of $^3C^*$ species that oxidize syringol were quantified in this study (Kaur and Anastasio, 2018).

The syringol decays followed pseudo first order kinetics (Figure S5). The syringol decay rates were used to calculate the

steady-state concentrations of $^3C^*$ ($[^3C^*]_{ss}$) as follows:

$$[^3C^*]_{ss} = \frac{1}{4} \sum_{i=1}^{i=4} \frac{k'_{SYR} - k^{SYR+^1O_2}_{rxn} \times [^1O_2]_{ss} - j_{SYR}}{k^{SYR+model\,^3C^*_i}_{rxn}} \qquad (8)$$

where $k'_{SYR}$ is the pseudo first order rate constant of syringol loss determined from the slope of the linear plot of $\ln([SYR]_t/[SYR]_0)$ vs. irradiation time (Figure S5), $k^{SYR+^1O_2}_{rxn}$ is the second order rate constant between syringol and $^1O_2$ (($3.6 \pm 0.7) \times 10^7$ $M^{-1}$ $s^{-1}$) (Tratnyek and Hoigne, 1991), $j_{SYR}$ is the first order direct photolysis rate of syringol ($2.62 \pm 0.12 \times 10^{-6}$ $s^{-1}$, Figure





S3), and $k_{\text{rxn}}^{\text{SYR+model }^3\text{C}_i^*}$ is the second order rate constant between syringol and a model $^3\text{C}^*$ species (Table S4). Since $^3\text{C}^*$ comprises of a variety of species with a range of reactivities, there is no single value for the rate constant of syringol with $^3\text{C}^*$. Thus, we used the method previously described by Kaur and Anastasio (2018) where an average of the rate constants for four model $^3\text{C}^*$ species (2-acetonaphthone ($^3\text{2AN}^*$), 3'-methoxyacetophenone ($^3\text{3MAP}^*$), 3,4-dimethoxybenzaldehyde ($^3\text{DMB}^*$), and benzophenone ($^3\text{BP}^*$)) was used to cover the range of $^3\text{C}^*$ reactivities in atmospheric samples. While previous studies performed ·OH photochemical experiments to correct for the reaction between syringol and ·OH in their $[^3\text{C}^*]_{\text{ss}}$ calculations (Kaur and Anastasio, 2018; Kaur et al., 2019; Ma et al., 2023b), we did not do so in our study due to insufficient extract volumes for additional photochemical experiments. However, previous studies have reported that the contribution of both ·OH and $^1\text{O}_2$ to the loss of syringol were < 20 % for the measurement of $[^3\text{C}^*]_{\text{ss}}$ in fog water (Kaur and Anastasio, 2018) and PM$_{2.5}$ extracts (Kaur et al., 2019).

The formation rate of $^3\text{C}^*$ ($R_{\text{f},^3\text{C}^*}$) was calculated as follows:

$$R_{\text{f},^3\text{C}^*} = [^3\text{C}^*]_{\text{ss}} \times (k_{\text{q},\text{O}_2}[\text{O}_2(\text{aq})] + k_{\text{rxn+q}}^{^3\text{C}^*+\text{WSOC}}[\text{WSOC}]) \tag{9}$$

where $k_{\text{q},\text{O}_2}$ is the average second order rate constant for the four model $^3\text{C}^*$ species quenching via energy transfer to dissolved $\text{O}_2$ ($2.8 \times 10^9 \text{ M}^{-1} \text{ s}^{-1}$) (Canonica et al., 2000; Kaur and Anastasio, 2018), $[\text{O}_2 \text{ (aq)}]$ is the dissolved $\text{O}_2$ concentration in water at 26 °C ($2.53 \times 10^{-4}$ M) (Rounds et al., 2006), $k_{\text{rxn+q}}^{^3\text{C}^*+\text{WSOC}}$ is the estimated overall rate constant for $^3\text{C}^*$ loss (i.e., reaction and quenching) due to WSOC ($9.3 \times 10^7 \text{ L mol-C}^{-1} \text{ s}^{-1}$) (Kaur et al., 2019), and [WSOC] (in mg-C L$^{-1}$) is the concentration of WSOC in each extract (Table S2).

The apparent quantum yield of $^3\text{C}^*$ ($\Phi_{^3\text{C}^*}$) was calculated as follows:

$$\Phi_{^3\text{C}^*} = \frac{R_{\text{f},^3\text{C}^*}}{R_{\text{abs}}} \tag{10}$$

Uncertainties were propagated from the measured decay kinetics of furfuryl alcohol and syringol in triplicate photochemical experiments and one standard deviation of the literature second order rate constants. Statistics and linear regression analyses were performed using Prism 8.

## 3 Results and discussion

### 3.1 Characteristics of the extracts

#### 3.1.1 WSOC and light absorption properties

The WSOC concentration in the PM$_{2.5}$ extracts ranged from 3.8 to 25.7 mg-C L$^{-1}$, with a study average of 13.7 mg-C L$^{-1}$ (Table S2). Converted to the carbon mass concentration in air, the study average WSOC concentration ($1.7 \pm 0.8$ µg m$^{-3}$) was close to previously reported values at another Hong Kong urban site ($1.8 \pm 1.1$ µg m$^{-3}$) and the semi-rural site HT ($1.3 \pm 1.1$ µg m$^{-3}$) in PM$_{2.5}$ (Huang et al., 2014). The WSOC concentration had a noticeable seasonal trend, wherein the concentrations were higher in the fall and winter extracts and the lowest concentrations were measured in the summer extracts (Table S2).



The seasonal variations in the WSOC concentration in PM$_{2.5}$ could be attributed to the seasonal variations in long-range air mass transport influenced by the East Asian monsoon system (Huang et al., 2014; Zhang et al., 2018; Chow et al., 2022). Air masses originating mainly from polluted continental areas located north of Hong Kong contributed to the high WSOC concentrations in fall and winter PM$_{2.5}$, whereas air masses originating mainly from clean marine regions located south of Hong Kong contributed to the low WSOC concentrations in summer PM$_{2.5}$ (Figures S6 to S8).

All the extracts had absorbance from the near-UV to the visible region, indicating the presence of BrC and the potential of generating $^1O_2$ and $^3C^*$ in all extracts. The absorption coefficient, $\alpha_\lambda$, and mass absorption coefficient, MAC$_\lambda$, declined exponentially with $\lambda$ for all the extracts (Figure 2). The average values of the absorption coefficient and mass absorption coefficient at 300 nm ($\alpha_{300}$ and MAC$_{300}$) indicated that, on average, the absorbance for the urban CU and TW extracts were slightly higher than the absorbance for the semi-rural HT extracts (Table 1). Upon grouping the $\alpha_{300}$ and MAC$_{300}$ datasets

based on seasonality irrespective of the sampling location, we observed noticeable differences in the seasonal $\alpha_{300}$ and MAC$_{300}$ values (Table 2). The average seasonal $\alpha_{300}$ and MAC$_{300}$ values followed similar trends: winter > fall > spring > summer. Since the MAC$_{300}$ accounts for WSOC dilution (Equation 3), the higher MAC$_{300}$ values in the winter extracts indicated that the water-soluble organic compounds in winter PM$_{2.5}$ were more strongly absorbing and/or were less diluted with weakly absorbing water-soluble organic compounds compared to the PM$_{2.5}$ from the other three seasons.

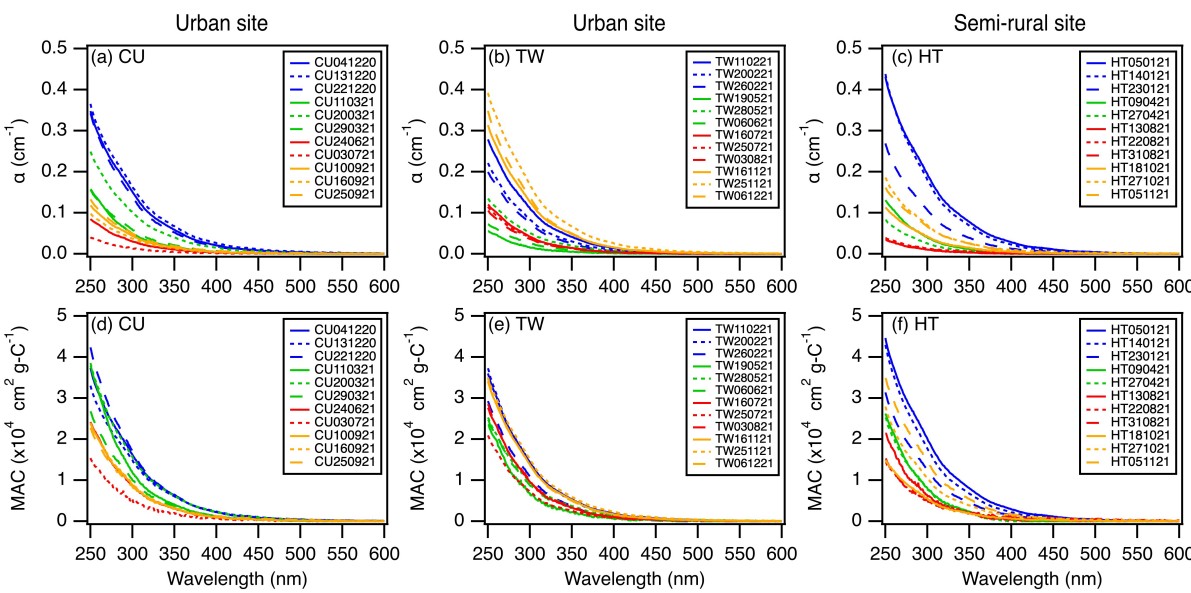

**Figure 2.** (a to c) $\alpha_\lambda$ and (d to f) MAC$_\lambda$ of PM$_{2.5}$ extracts from CU, TW, and HT, respectively. The lines in blue, green, red, and orange indicated samples collected during the winter, spring, summer, and fall seasons, respectively.

The AAE describes the spectral dependence of light absorption, and is typically used to indicate BrC contribution to the total absorption of aerosols (Helin et al., 2021). The AAE value for black carbon is typically close to 1, while AAE values larger than 1 indicate the presence of BrC (Kirchstetter et al., 2004). All the AAE values were larger than 1, thus indicating



the omnipresence of BrC. The AAE values were fairly similar among the three sites (Table 1) and across the four seasons (Table 2). The $R_{abs}$ values summarize the light absorption rates ranging from 290 to 600 nm. $R_{abs}$ was linearly correlated with

the WSOC concentration, with Pearson's $r$ values between 0.88 and 0.97 for the three sites (Figure S9). The good correlation between $R_{abs}$ and the WSOC concentration implied that water-soluble BrC was likely the main contributor to the total light absorption.

$SUVA_{254}$ and $SUVA_{365}$, which are the specific UV absorbance obtained from dividing the absorption coefficients at 254 nm and at 365 nm by the WSOC concentration, are commonly used as proxies for organic matter aromaticity. Higher $SUVA_{254}$

and $SUVA_{365}$ values indicate enhanced aromaticity (Weishaar et al., 2003). As expected, the $SUVA_{254}$ values for the three sites were higher than the $SUVA_{365}$ values. These average $SUVA_{254}$ and $SUVA_{365}$ values for the three sites indicated that the organic matter in the urban CU and TW extracts, on average, had higher aromaticity than those in the semi-rural HT extracts (Table 1). It is possible that the observed higher absorbance and aromaticity in the urban CU and TW extracts were due to the presence of aromatic compounds (e.g., polycyclic aromatic hydrocarbons) from local vehicle emissions and other

combustion-related activities (Kuang et al., 2018; Wong et al., 2020). Upon grouping the $SUVA_{254}$ and $SUVA_{365}$ datasets based on seasonality irrespective of the sampling location, the average seasonal $SUVA_{254}$ and $SUVA_{365}$ values indicated that the organic matter in the fall and winter extracts, on average, had higher aromaticity than those in the spring and summer extracts (Table 2).

### 3.1.2 Site and seasonal variations in WSOC and light absorption properties

We hypothesized that the site and seasonal variations in the WSOC concentration and light absorption properties of water-soluble BrC in the $PM_{2.5}$ drove the site and seasonal variations in $^3C^*$ and $^1O_2$ production. Thus, we examined the site and seasonal variations in the WSOC and light absorption properties of the extracts. The above comparisons of the average WSOC concentration, $\alpha_{300}$, $MAC_{300}$, $SUVA_{254}$, and $SUVA_{365}$ values of the urban CU and TW extracts vs. semi-rural HT extracts indicated that, on average, $PM_{2.5}$ at CU and TW had slightly higher concentrations of and/or more absorbing water-soluble

BrC comprised of organic matter of high aromaticity compared to $PM_{2.5}$ at HT. However, statistics performed on the WSOC concentration, $\alpha_{300}$, $MAC_{300}$, AAE, $R_{abs}$, $SUVA_{254}$, and $SUVA_{365}$ datasets showed that their variations between the three sites were not significant ($p > 0.05$) (Table S3). These results implied that the water-soluble BrC in $PM_{2.5}$ was weakly influenced by local emission sources near the sites.

Since the locations (urban vs. semi-rural) did not have a significant influence on the the WSOC concentration and light ab-

sorption properties of water-soluble BrC in the extracts, we combined the datasets from the three sites and separated them based on seasonality. Despite the spread in their seasonal values, seasonal variations in the WSOC concentration, $\alpha_{300}$, $MAC_{300}$, $R_{abs}$, $SUVA_{254}$, and $SUVA_{365}$ values were statistically significant ($p < 0.05$) (Figure S10). This implied that the seasonal variations in long-range air mass transport had a significant influence on the WSOC concentration and light absorption properties of water-soluble BrC. The WSOC concentration, $\alpha_{300}$, $MAC_{300}$, $R_{abs}$, $SUVA_{254}$, and $SUVA_{365}$ had noticeably similar

trends: winter > fall > spring > summer. These seasonal trends indicated that winter and fall $PM_{2.5}$ had higher concentrations of and/or more absorbing water-soluble BrC comprised of organic matter of high aromaticity compared to the summer and



**Table 1.** Summary of WSOC concentration, optical parameters, steady-sate concentrations, and quantum yields of $^1O_2$ and $^3C^*$ for the CU, TW, and HT sites.

| Parameters | Units | CU | | TW | | HT | |
|---|---|---|---|---|---|---|---|
| | | Range | Average | Range | Average | Range | Average |
| [WSOC] | mg-C L$^{-1}$ | 6.03-25.46 | 13.79 ± 5.96 | 5.41-25.73 | 14.38 ± 6.51 | 3.76-23.46 | 13.01 ± 7.05 |
| $\alpha_{300}$ | cm$^{-1}$ | 0.014-0.164 | 0.076 ± 0.053 | 0.016-0.169 | 0.076 ± 0.050 | 0.010-0.194 | 0.069 ± 0.065 |
| $R_{abs}$ | ×10$^{-6}$ mol-photons L$^{-1}$ s$^{-1}$ | 0.37-5.23 | 2.23 ± 1.66 | 0.39-5.17 | 2.11 ± 1.47 | 0.20-5.82 | 1.91 ± 1.93 |
| MAC$_{300}$ | ×10$^4$ cm$^2$ g-C$^{-1}$ | 0.52-1.49 | 1.15 ± 0.39 | 0.68-1.51 | 1.12 ± 0.29 | 0.64-2.01 | 1.03 ± 0.53 |
| SUVA$_{254}$ | L mg-C$^{-1}$ m$^{-1}$ | 0.613-1.327 | 1.190 ± 0.345 | 0.951-1.426 | 1.201 ± 0.224 | 0.854-1.798 | 1.087 ± 0.428 |
| SUVA$_{365}$ | L mg-C$^{-1}$ m$^{-1}$ | 0.054-0.195 | 0.134 ± 0.053 | 0.068-0.192 | 0.125 ± 0.039 | 0.051-0.256 | 0.113 ± 0.071 |
| AAE | | 6.45-8.30 | 7.38 ± 0.56 | 6.82-8.07 | 7.40 ± 0.45 | 5.31-8.56 | 7.19 ± 1.01 |
| [$^1O_2$]$_{ss}$ | ×10$^{-13}$ M | 0.16-8.22 | 3.41 ± 2.54 | 0.33-8.88 | 4.30 ± 2.97 | 0.23-13.47 | 4.32 ± 4.93 |
| $\Phi_{^1O_2}$ | % | 1.19-7.21 | 4.31 ± 1.70 | 2.11-9.54 | 5.60 ± 2.31 | 1.35-13.74 | 5.62 ± 3.58 |
| [$^3C^*$]$_{ss}$ | ×10$^{-15}$ M | 1.41-17.19 | 9.59 ± 5.16 | 0.35-27.27 | 7.76 ± 7.19 | 0.29-80.77 | 15.73 ± 22.62 |
| $\Phi_{^3C^*}$ | % | 0.18-0.85 | 0.44 ± 0.26 | 0.06-2.28 | 0.47 ± 0.64 | 0.05-3.24 | 0.77 ± 0.91 |

spring PM$_{2.5}$. Based on the seasonal variations in long-range air mass transport during the study (Figures S6 to S8), regional sources were important contributors to water-soluble BrC comprised of organic matter of high aromaticity in winter and fall PM$_{2.5}$. Interestingly, seasonal variations in the AAE values were not statistically significant ($p > 0.05$). While it is unclear why seasonal trends were not observed for the AAE values in our study, other studies have similarly reported the lack of seasonal trends in the AAE values (Du et al., 2014; Ma et al., 2023a).

### 3.2 $^1O_2$ and $^3C^*$ production during extract illumination

#### 3.2.1 $^1O_2$

Since the $^1O_2$ measurements were used to determine $^3C^*$ production, we present the $^1O_2$ measurements first. The pseudo-first order decay rate constants of furfuryl alcohol ($^1O_2$ chemical probe) in photochemical experiments (Figure S4) were used to determine the steady-state concentrations of $^1O_2$, [$^1O_2$]$_{ss}$ (Equation 5). The [$^1O_2$]$_{ss}$ values spanned two orders of magnitude, ranging from $1.56 \times 10^{-14}$ to $1.35 \times 10^{-12}$ M, with a study average of $(4.02 \pm 3.52) \times 10^{-13}$ M (Table S5). The range of [$^1O_2$]$_{ss}$ values is remarkably large, and although others have reported $^1O_2$ values between $10^{-15}$ to $10^{-12}$ M, they have not been within the same study (Table S7). The [$^1O_2$]$_{ss}$ values were linearly correlated with two indicators of water-soluble BrC, WSOC concentration and $\alpha_{300}$, with Pearson's $r$ values of 0.88 and 0.92, respectively (Figures S11a and S11b). These correlations provided strong evidence that the production of $^1O_2$ was linked to water-soluble BrC. The large range in the [$^1O_2$]$_{ss}$ values was likely due to the variations in the quantity and absorbance of the BrC chromophores.



**Table 2.** Summary of WSOC concentration, optical parameters, steady-sate concentrations, and quantum yields of $^1O_2$ and $^3C^*$ for the four seasons.

| Parameters | Winter | | Spring | | Summer | | Fall | |
|---|---|---|---|---|---|---|---|---|
| | Range | Average | Range | Average | Range | Average | Range | Average |
| [WSOC] | 13.66-25.46 | $19.80 \pm 3.77$ | 5.41-14.90 | $10.15 \pm 3.40$ | 3.76-12.59 | $7.54 \pm 2.84$ | 9.45-25.73 | $16.41 \pm 5.83$ |
| $\alpha_{300}$ | 0.075-0.194 | $0.134 \pm 0.042$ | 0.016-0.101 | $0.046 \pm 0.027$ | 0.010-0.042 | $0.025 \pm 0.014$ | 0.037-0.169 | $0.083 \pm 0.049$ |
| $R_{abs}$ | 1.72-5.82 | $3.83 \pm 1.40$ | 0.33-3.14 | $1.23 \pm 0.92$ | 0.20-1.08 | $0.68 \pm 0.34$ | 1.01-5.17 | $2.35 \pm 1.45$ |
| $MAC_{300}$ | 1.10-2.01 | $1.53 \pm 0.28$ | 0.68-1.56 | $0.99 \pm 0.28$ | 0.49-1.00 | $0.72 \pm 0.21$ | 0.55-1.51 | $1.11 \pm 0.33$ |
| $SUVA_{254}$ | 1.183-1.798 | $1.493 \pm 0.215$ | 0.951-1.550 | $1.146 \pm 0.244$ | 0.578-1.142 | $0.840 \pm 0.228$ | 0.597-1.426 | $1.126 \pm 0.293$ |
| $SUVA_{365}$ | 0.106-0.256 | $0.181 \pm 0.044$ | 0.045-0.197 | $0.104 \pm 0.047$ | 0.051-0.101 | $0.077 \pm 0.020$ | 0.059-0.192 | $0.127 \pm 0.042$ |
| AAE | 6.66-7.80 | $7.32 \pm 0.36$ | 6.86-8.56 | $7.51 \pm 0.65$ | 5.31-8.07 | $6.98 \pm 1.08$ | 6.45-7.89 | $7.41 \pm 0.56$ |
| $[^1O_2]_{ss}$ | 4.92-13.47 | $7.58 \pm 2.67$ | 0.33-2.73 | $1.60 \pm 0.90$ | 0.16-3.14 | $1.15 \pm 1.18$ | 1.38-10.08 | $5.15 \pm 3.52$ |
| $\Phi_{^1O_2}$ | 3.49-8.78 | $5.92 \pm 1.82$ | 2.24-6.47 | $4.07 \pm 1.40$ | 1.19-9.54 | $4.36 \pm 3.28$ | 2.62-13.74 | $6.19 \pm 3.22$ |
| $[^3C^*]_{ss}$ | 2.41-17.90 | $11.08 \pm 6.50$ | 0.35-15.80 | $6.44 \pm 4.31$ | 0.29-27.27 | $7.62 \pm 9.47$ | 2.97-80.77 | $17.72 \pm 23.95$ |
| $\Phi_{^3C^*}$ | 0.06-0.49 | $0.24 \pm 0.13$ | 0.07-1.44 | $0.50 \pm 0.40$ | 0.05-2.28 | $0.69 \pm 0.74$ | 0.14-3.24 | $0.80 \pm 0.98$ |

Note: The unit for each parameter is the same as in Table 1.

The $[^1O_2]_{ss}$ values were used to determine the formation rates of $^1O_2$ , $R_{f,^1O_2}$ (Equation 6). The $R_{f,^1O_2}$ values ranged from $4.39 \times 10^{-9}$ to $3.79 \times 10^{-7}$ M s$^{-1}$ (Table S5). Across all extracts, the $R_{f,^1O_2}$ was linearly correlated with $R_{abs}$ (Figure S12a), which was consistent with water-soluble BrC being a source of $^1O_2$ . The quantum yields of $^1O_2$ , $\Phi_{^1O_2}$, which can be viewed as an indicator of the $^1O_2$ photosensitization efficiency, was subsequently calculated from the $R_{f,^1O_2}$ values (Equation 7). The $\Phi_{^1O_2}$ values ranged from 0.77 to 13.74 %. The study's average $\Phi_{^1O_2}$ was $(5.12 \pm 2.66)$ %, which was noticeably higher than previously reported $\Phi_{^1O_2}$ values for atmospheric PM samples (0.3 to 4.5 %) (Kaur and Anastasio, 2017; Manfrin et al., 2019; Kaur et al., 2019; Leresche et al., 2021; Bogler et al., 2022). This suggested that the water-soluble BrC in our study's extracts have higher $^1O_2$ photosensitization efficiencies compared to the atmospheric PM samples investigated in previous studies, which could be due to the different composition and age of water-soluble BrC in atmospheric PM in different locations. However, we cannot discount the possibility that the higher $\Phi_{^1O_2}$ values observed in our study could be due to differences in experimental conditions. For instance, we used UVA light to illuminate the extracts in photochemical experiments, whereas previous studies used xenon arc lamps (Kaur et al., 2019) or a solar simulator instrument (Leresche et al., 2021). In addition, the different methodologies used to determine $\Phi_{^1O_2}$ may have contributed to our study's higher $\Phi_{^1O_2}$ values (Manfrin et al., 2019; Bogler et al., 2022)

### 3.2.2  $^3C^*$

Table S6 summarizes the $^3C^*$ measurements. The pseudo-first order decay rate constants of syringol ($^3C^*$ chemical probe) in photochemical experiments (Figure S5) were used to determine the steady-state concentrations of $^3C^*$ , $[^3C^*]_{ss}$ (Equation 8).



However, it is important to note that due to the chemical complexity of $^3C^*$ species, a single chemical probe cannot quantify all the $^3C^*$ species (Maizel and Remucal, 2017). Hence, only a subset of $^3C^*$ species that can oxidize syringol was quantified in our study (Kaur and Anastasio, 2018). The $[^3C^*]_{ss}$ values spanned two orders of magnitude, ranging from $2.93 \times 10^{-16}$ to $8.08 \times 10^{-14}$ M, with a study average of $(1.09 \pm 1.39) \times 10^{-14}$ M. While the range of $[^3C^*]_{ss}$ values was in line with those previously measured in atmospheric samples ($10^{-16}$ to $10^{-12}$ M) (Table S7), not all of these previous studies used syringol as

the $^3C^*$ chemical probe. The choice of the $^3C^*$ chemical probe can impact the $[^3C^*]_{ss}$ measurements. This is because different $^3C^*$ chemical probes react with different subsets of $^3C^*$ species of different oxidizing abilities (Kaur and Anastasio, 2018; Ma et al., 2023b). In addition, the $^3C^*$ chemical probes may be inhibited by the copresence of some atmospheric species (e.g., copper, water-soluble organic matter), albeit by different extents (Ma et al., 2023b). Nevertheless, the $[^3C^*]_{ss}$ values were linearly correlated with the WSOC concentration and $\alpha_{300}$ (Figures S11c and S11d), which was consistent with water-soluble

BrC being a source of $^3C^*$. The correlations of $[^3C^*]_{ss}$ with the WSOC concentration and $\alpha_{300}$ were noticeably weaker than the correlations of $[^1O_2]_{ss}$ with the WSOC concentration and $\alpha_{300}$. The weaker $[^3C^*]_{ss}$ correlations could be attributed to the chemical complexity of the $^3C^*$ pool. Even though water-soluble BrC is a key precursor of $^3C^*$, the sample-to-sample variability in the subset of $^3C^*$ species that were able to oxidize syringol likely caused the weaker $[^3C^*]_{ss}$ correlations with the WSOC concentration and $\alpha_{300}$.

The $[^3C^*]_{ss}$ values were used to determine the formation rates of $^3C^*$, $R_{f,^3C^*}$ (Equation 9). The $R_{f,^3C^*}$ values ranged from $2.20 \times 10^{-10}$ to $6.68 \times 10^{-8}$ M s$^{-1}$, with a study average of $(9.07 \pm 11.50) \times 10^{-9}$ M s$^{-1}$ (Table S6). Across all extracts, the $R_{f,^3C^*}$ was linearly correlated with $R_{abs}$, with a Pearson's *r* value of 0.63 (Figure S12b), which indicated that $^3C^*$ production was linked to water-soluble BrC. The correlation between $R_{f,^3C^*}$ and $R_{abs}$ was weaker than the correlation between $R_{f,^1O_2}$ and $R_{abs}$. Kaur et al. (2019) similarly reported weaker linear correlations for $R_{abs}$ vs. $R_{f,^3C^*}$ compared to $R_{abs}$ vs. $R_{f,^1O_2}$ for

extracts of winter PM$_{2.5}$ collected from areas influenced by biomass burning emissions in California, USA. Sample-to-sample variability in the subset of $^3C^*$ species that were able to oxidize syringol likely caused the weaker $R_{abs}$ vs. $R_{f,^3C^*}$ correlations. On average, $R_{abs}$ was about 20 times higher than the sum of $R_{f,^1O_2}$ and $R_{f,^3C^*}$. This indicated that majority of the (photo) energy absorbed by the illuminated extracts in the photochemical experiments were dissipated by non-reactive pathways and/or led to the formation of products other than $^1O_2$ and/or $^3C^*$.

The quantum yields of $^3C^*$, $\Phi_{^3C^*}$, which can be viewed as an indicator of the $^3C^*$ photosensitization efficiency, were subsequently calculated from the $R_{f,^3C^*}$ values (Equation 10). The $\Phi_{^3C^*}$ values ranged from 0.05 to 3.24 %. The study average $\Phi_{^3C^*}$ was $(0.55 \pm 0.66)$ %, which was approximately 9 times lower than the study average $\Phi_{^1O_2}$. The difference in $^3C^*$ and $^1O_2$ photosensitization efficiencies could be due to only a subset of $^3C^*$ species that can oxidize syringol being captured in our photochemical experiments since different $^3C^*$ species may have different photosensitization efficiencies. Our study average

$\Phi_{^3C^*}$ was also lower than the average $\Phi_{^3C^*}$ $((2.40 \pm 1.00)$ %) reported by Kaur et al. (2019) for extracts of PM collected from biomass burning-influenced areas in California, USA. This suggested that the water-soluble BrC in our study's extracts have lower $^3C^*$ photosensitization efficiencies compared to the water-soluble BrC in PM samples investigated by Kaur et al. (2019), which could be due to the different composition and age of water-soluble BrC in atmospheric PM in different locations. However, we cannot discount the possibility that the lower $\Phi_{^3C^*}$ values observed in our study could also be due, in part, to





the differences in experimental conditions and methodology. For instance, we used UVA light to illuminate the extracts and syringol as the sole $^3C^*$ chemical probe in photochemical experiments, whereas Kaur et al. (2019) used a xenon arc lamp to illuminate their extracts and syringol and methyl jasomonate as $^3C^*$ chemical probes in their photochemical experiments.

### 3.3   Site and seasonal variations of $^1O_2$ and $^3C^*$ production

Both the $[^1O_2]_{ss}$ and $[^3C^*]_{ss}$ values were fairly similar among the three sites (Table 1). Due to the large spreads in the $[^1O_2]_{ss}$
and $[^3C^*]_{ss}$ values for each site, the $[^1O_2]_{ss}$ and $[^3C^*]_{ss}$ values did not vary significantly between the three sites ($p > 0.05$) (Figures S13a and S13b). Similarly, both the $\Phi_{^1O_2}$ and $\Phi_{^3C^*}$ values were fairly similar among the three sites (Figures S13c and S13d). Variations in $\Phi_{^1O_2}$ and $\Phi_{^3C^*}$ across the three sites were also not statistically significant ($p > 0.05$) (Figures S13c and S13d). Taken together, this indicated that the location (i.e., urban vs. semi-rural) did not have a significant effect on the steady-state concentrations and photosensitization efficiencies of $^1O_2$ and $^3C^*$, which implied that BrC from local $PM_{2.5}$ sources did
not have a significant influence on the year-round $^3C^*$ and $^1O_2$ production. The large spreads in the steady-state concentrations and quantum yields of $^1O_2$ and $^3C^*$ for the three sites highlighted the broad range of BrC chromophores present in the $PM_{2.5}$ at the three locations that are capable of photosensitizing $^1O_2$ and $^3C^*$.

Since the locations did not have a significant influence on the the steady-state concentrations and quantum yields of $^1O_2$ and $^3C^*$, we combined the $^1O_2$ and $^3C^*$ datasets from the three sites and separated them based on seasonality. We observed a
distinct seasonal trend for $[^1O_2]_{ss}$ (Figure 3a). The $[^1O_2]_{ss}$ values were generally the highest in the winter, and the lowest in the summer (Table 2). The seasonal variations in the $[^1O_2]_{ss}$ were also found to be statistically significant ($p < 0.05$). The seasonal trend for $[^3C^*]_{ss}$ was noticeably weaker and was not statistically significant ($p > 0.05$) (Figure 3b). However, the $[^3C^*]_{ss}$ values were mostly higher in the fall and winter and lower in the spring and summer (Table 2). The differences in the strengths of the seasonal trends of $[^1O_2]_{ss}$ (i.e., strong and statistically significant) and $[^3C^*]_{ss}$ (i.e., weak and statistically insignificant)
could be attributed to sample-to-sample variations in $^3C^*$ species that can form $^1O_2$. Even though $^3C^*$ is a precursor of $^1O_2$, not all $^3C^*$ species will form $^1O_2$. In addition, high energy and strongly reducing $^3C^*$ species are not necessarily efficient $^1O_2$ photosensitizers (McNeill and Canonica, 2016). Sample-to-sample variability in the subset of $^3C^*$ species that were able to oxidize syringol could also have contributed to the weak seasonal $[^3C^*]_{ss}$ trend. The fall $[^3C^*]_{ss}$ average (Table 2) was noticeably high, and this was due to the inclusion of an abnormally high $[^3C^*]_{ss}$ value (($8.08 \pm 4.59) \times 10^{-14}$ M) obtained for
the HT271021 sample which was identified as a "far out outlier" by Tukey's fences. Unlike the other samples, we observed fast photobleaching for the HT271021 sample during the photochemical experiments (Figures S4 and S5), which likely resulted in over-estimated steady-state concentrations (Sections 2.4 and 2.5). It should be noted that while a high $[^1O_2]_{ss}$ value was also obtained for the HT271021 sample, it was not identified as an outlier by Tukey's fences.

Overall, seasonality had a significant effect on the steady-state concentrations of $^1O_2$ and $^3C^*$, wherein $[^1O_2]_{ss}$ and $[^3C^*]_{ss}$
were the highest in the fall and winter and the lowest in the summer. Most importantly, the $[^1O_2]_{ss}$ and $[^3C^*]_{ss}$ seasonal trends correlated with the seasonal variations in the WSOC concentration and light absorption properties of water-soluble BrC (Figure S10). As discussed in Section 3.1.2, the winter and fall extracts had higher concentrations of and/or more absorbing water-soluble BrC comprised of organic matter of high aromaticity, whereas the converse was observed for summer and spring




extracts. Taken together, this indicated that the higher concentrations of and/or more absorbing water-soluble BrC in the winter

and fall extracts enhanced $^1O_2$ and $^3C^*$ production. Since the seasonal variations in the WSOC concentration and light absorption properties of water-soluble BrC were due to the seasonal variations in long-range air mass transport (Figures S6 to S8), this implied that regional $PM_{2.5}$ sources located in continental areas north of Hong Kong contributed to the higher production of $^1O_2$ and $^3C^*$ in the fall and winter.

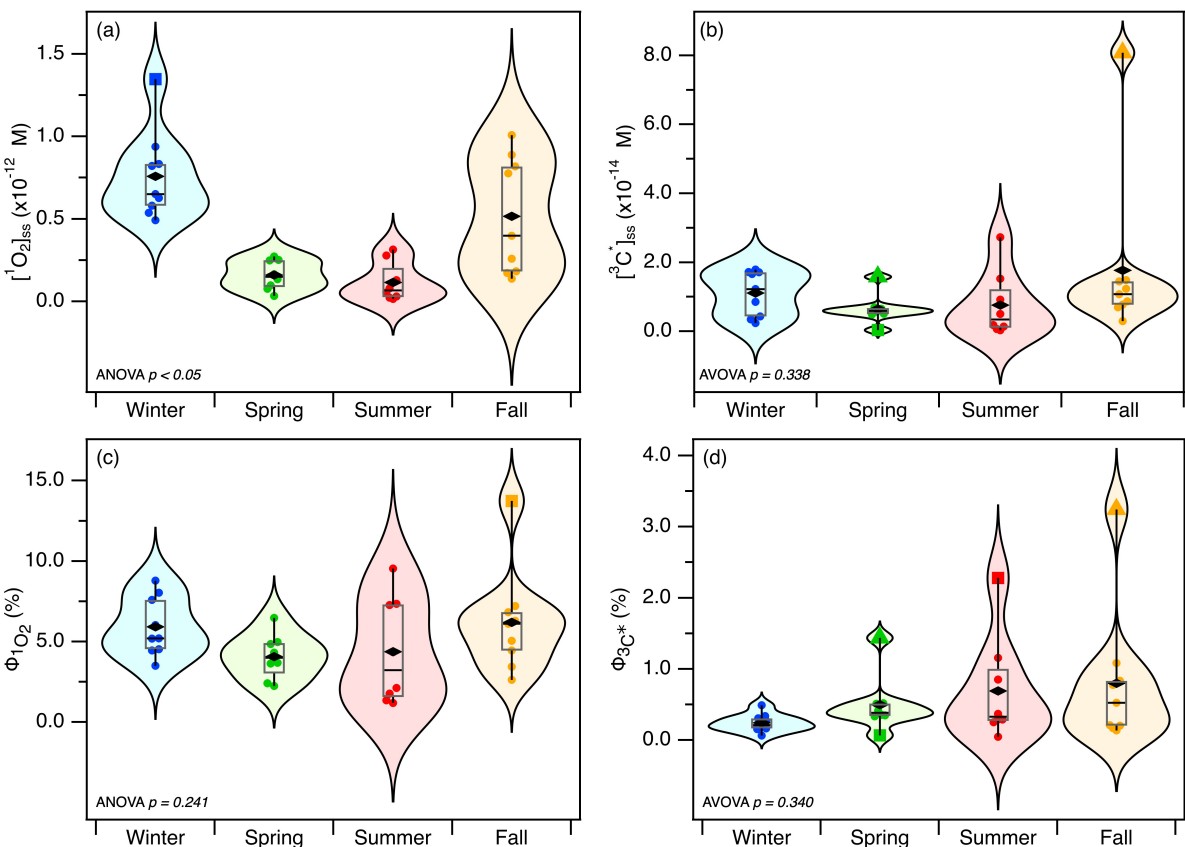

**Figure 3.** Violin plots showing the seasonal variations of (a) $[^1O_2]_{ss}$, (b) $[^3C^*]_{ss}$, (c) $\Phi_{^1O_2}$, and (d) $\Phi_{^3C^*}$. For the box plots, the squares indicate "far out outliers" and the triangles indicate outliers identified by Tukey's fences, the whiskers denote the minimum and maximum values, the boxes denote the $25^{th}$ and $75^{th}$ percentile values, black diamonds indicate the mean values, and the boxes' midline denote the median values.

The seasonal trends of $\Phi_{^1O_2}$ and $\Phi_{^3C^*}$ (Figures 3c and 3d) were noticeably weaker than the seasonal trends of $[^1O_2]_{ss}$ and

$[^3C^*]_{ss}$ (Figures 3a and 3b). The average $\Phi_{^1O_2}$ for the winter, spring, summer, and fall were (5.92 ± 1.82) %, (4.07 ± 1.40) %, (4.36 ± 3.28) %, and (6.19 ± 3.22) %, respectively. Even after accounting for their spread, the average seasonal $\Phi_{^1O_2}$ values suggested that the $^1O_2$ photosensitization efficiency was higher in the fall and winter. The average $\Phi_{^3C^*}$ for the winter, spring,





summer, and fall were $(0.24 \pm 1.23)$ %, $(0.50 \pm 0.40)$ %, $(0.69 \pm 0.74)$ %, and $(0.80 \pm 0.98)$ %, respectively. However, the average seasonal $\Phi_{^3C^*}$ values did not have an obvious seasonal trend due to their spread and standard deviations. The average

$\Phi_{^1O_2}$ and $\Phi_{^3C^*}$ values were noticeably the highest for the fall season. This was due to the inclusion of abnormally high quantum yield values obtained for the HT271021 sample (identified as a "far out outlier" by Tukey's fences). Fast photobleaching for the HT271021 sample was observed during the photochemical experiments (Figures S4 and S5), and this could have resulted in over-estimated quantum yields based on the methodology we used to calculate quantum yield values (Sections 2.4 and 2.5). Nevertheless, the variations in $\Phi_{^1O_2}$ and $\Phi_{^3C^*}$ across the four seasons were not statistically significant ($p > 0.05$). Taken

together, this indicated that seasonality did not have a significant effect on the photosensitization efficiencies of $^1O_2$ and $^3C^*$.

We also compared the influence of seasonal variations in long-range air mass transport on the $[^1O_2]_{ss}$ and $[^3C^*]_{ss}$ values for the urban CU and TW sites vs. semi-rural HT site. Since the spring sampling months could be viewed as a transition period wherein the dominant air masses that arrive in Hong Kong gradually shifted from the polluted continental northern areas (fall and winter months) to the clean marine southern regions (summer months) (Figures S6 to S8), for simplicity, we excluded

the spring datatsets from this comparison. The fall and winter datatsets were combined and the subsequent average value was compared to the average value of the summer datatset. Larger contrasts in the $[^1O_2]_{ss}$ and $[^3C^*]_{ss}$ values were observed for the semi-rural HT site compared to the urban CU and TW sites (Tables S5 and S6 ), which were line with the larger contrasts in the average WSOC concentrations and optical parameters for HT compared to CU and TW (Tables S2 and S3). This could be attributed to the nature of the sites. Due to the seasonal variations in long-range air mass transport (Figures S6 to S8), local

sources are the main contributors to summer $PM_{2.5}$, whereas regional sources located in continental areas north of Hong Kong are the main contributors to fall and winter $PM_{2.5}$ (Pathak et al., 2003; Louie et al., 2005; Louie, 2005; Huang et al., 2014; Li et al., 2015; Wong et al., 2020). In contrast to the urban CU and TW sites, the semi-rural HT site is located far from urban areas (approximately 6 km away from the nearest urban area). Thus, contributions of local anthropogenic emissions (e.g., traffic, combustion-related activities) to water-soluble BrC in summer $PM_{2.5}$ at the semi-rural HT site are smaller compared to those

at the urban CU and TW sites. This would result in larger contrasts between the average WSOC concentrations and optical parameters from the combined fall + winter datatset vs. summer dataset for the semi-rural HT site compared to the urban CU and TW sites. Consequently, the higher concentrations of water-soluble BrC in summer $PM_{2.5}$ from local anthropogenic emissions at the urban CU and TW sites contributed to their higher summer $[^1O_2]_{ss}$ and $[^3C^*]_{ss}$ values, and consequently smaller fall + winter vs. summer $[^1O_2]_{ss}$ and $[^3C^*]_{ss}$ contrasts, compared to semi-rural HT site.

### 3.4  Relating $[^3C^*]_{ss}$ and $[^1O_2]_{ss}$ to water-soluble BrC concentration and light absorption properties

To examine more closely how water-soluble BrC contributed to $^1O_2$ and $^3C^*$ production, we first investigated how the $[^1O_2]_{ss}$ and $[^3C^*]_{ss}$ values changed as a function of $MAC_{300}$, a light absorbance parameter that accounts for WSOC dilution. Both $[^1O_2]_{ss}$ and $[^3C^*]_{ss}$ showed positive correlations with $MAC_{300}$ (Figures 4a and 4c), which indicated that the production of $^1O_2$ and $^3C^*$ were governed by the quantity and absorbance of water-soluble BrC. $[^1O_2]_{ss}$ was noticeably more strongly linearly

correlated with $MAC_{300}$ compared to $[^3C^*]_{ss}$. The weaker $[^3C^*]_{ss}$ correlations could be attributed to the chemical complexity of the $^3C^*$ pool which cannot be quantified completely by syringol. Thus, even though water-soluble BrC is a key precursor




of $^3C^*$, the sample-to-sample variability in the size of the population of $^3C^*$ species that were able to oxidize syringol likely caused the weaker $[^3C^*]_{ss}$ correlations with $MAC_{300}$.

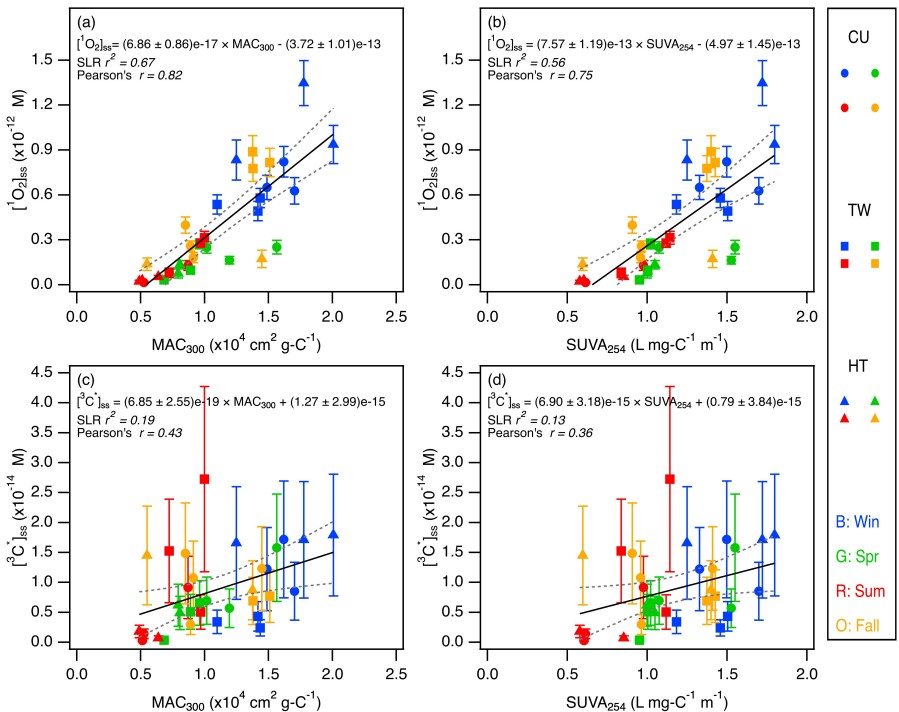

**Figure 4.** (a and b) $[^1O_2]_{ss}$ and (c and d) $[^3C^*]_{ss}$ as a function of $MAC_{300}$ and $SUVA_{254}$. The outlier, HT271021, was excluded. Blue, green, red, and orange symbols denote the winter, spring, summer, and fall samples, respectively. Dashed lines represent 95 % confidence bands. SLR $r^2$ and Pearson's $r$ indicate coefficient of determination of simple linear regression and Pearson correlation coefficient, respectively.

The $[^1O_2]_{ss}$ and $[^3C^*]_{ss}$ depend on both the quality and quantity of the BrC chromophores. The quantity of the BrC chro-
mophores is associated with their concentrations, whereas the quality of the BrC chromophores is linked to the specific ab-
sorbance of the BrC chromophores present (Bogler et al., 2022). A high WSOC concentration in an extract will result in a high
$[^1O_2]_{ss}$ (and/or a high $[^3C^*]_{ss}$) only if a high concentration of water-soluble BrC chromophores is present in the extract. The
relative importance in the quantity vs. quality of BrC chromophores in our study could be ascertained from the comparison
of the seasonal trends of $[^1O_2]_{ss}$ and $[^3C^*]_{ss}$ (Figures 3a and 3b) vs. the seasonal trends of $\Phi_{^1O_2}$ and $\Phi_{^3C^*}$ (Figures 3c and
3d). Stronger seasonal trends were observed for $[^1O_2]_{ss}$ and $[^3C^*]_{ss}$, which suggested that the quantity of BrC chromophores
mainly governed $^1O_2$ and $^3C^*$ production in our study.

We also normalized the $[^1O_2]_{ss}$ and $[^3C^*]_{ss}$ values determined for each extract by their WSOC concentrations and compared
the resulting seasonal variations (Figures 5a and 5b) to the seasonal trends for the unnormalized $[^1O_2]_{ss}$ and $[^3C^*]_{ss}$ (Figures
3a and 3b). A similar, albeit weaker, seasonal trend for the normalized $[^1O_2]_{ss}$ (Figure 5a) was observed compared to the
unnormalized $[^1O_2]_{ss}$ (Figure 3a). For both the normalized and unnormalized $[^1O_2]_{ss}$, the highest and lowest seasonal average



values were obtained for winter and summer, respectively. The ratio of the average normalized $[^1O_2]_{ss}$ for winter vs. summer was 2.68, which was substantially smaller than the the ratio of the average unnormalized $[^1O_2]_{ss}$ for winter vs. summer (6.59). In the case of $^3C^*$ , a weak (and statistically insignificant) seasonal trend was observed for the unnormalized $[^3C^*]_{ss}$, wherein the highest and lowest seasonal average values were obtained for winter and spring, respectively. The ratio of the average

unnormalized $[^3C^*]_{ss}$ for winter vs. spring was 1.72 (Figure 3b), which was larger than the ratio of the average normalized $[^3C^*]_{ss}$ for winter vs. spring (0.89) (Figure 5b). Taken together, the weakened seasonal trends for the $[^1O_2]_{ss}$ and $[^3C^*]_{ss}$ values upon normalization to the WSOC concentrations underscored the key role that BrC chromophore quantity plays in driving $^1O_2$ and $^3C^*$ production in our study.

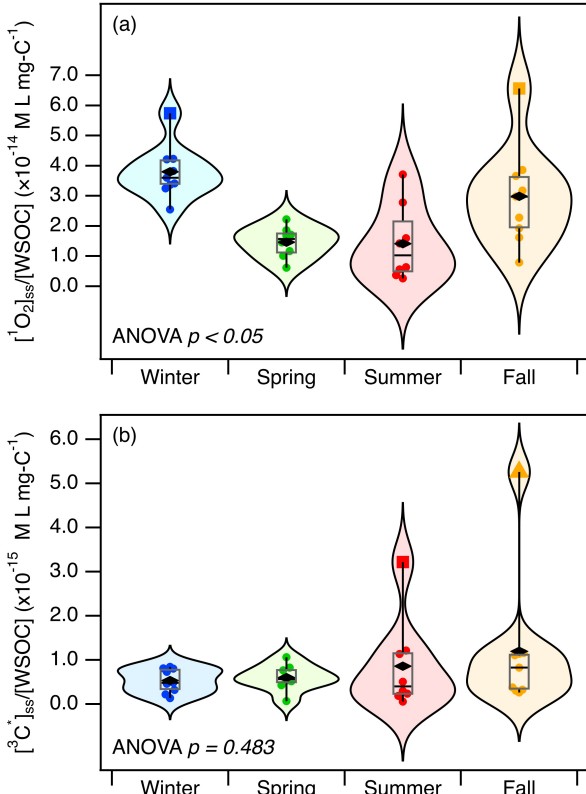

**Figure 5.** Violin plots showing the seasonal variations of WSOC normalized (a) $[^1O_2]_{ss}$ and (b) $[^3C^*]_{ss}$. For the box plots, the squares indicate "far out outliers" and the triangles indicate outliers identified by Tukey's fences, the whiskers denote the minimum and maximum values, the boxes denote the $25^{th}$ and $75^{th}$ percentile values, black diamonds indicate the mean values, and the boxes' midline denote the median values.

Even though the quantity of BrC chromophores appeared to the main driver of $^1O_2$ and $^3C^*$ production in our study, it is

still worth investigating factors that affected the quality of BrC chromophores. We hypothesized that the quality of BrC chro-



mophores was influenced by the presence of light absorbing aromatic compounds (Laskin et al., 2015). To test this hypothesis, we evaluated the contributions of aromatic compounds to $^1O_2$ and $^3C^*$ production by plotting the $[^1O_2]_{ss}$ and $[^3C^*]_{ss}$ values as a function of two commonly used indicators of aromaticity, $SUVA_{254}$ and $SUVA_{365}$ (Figures 4b, 4d, and S14). Both $[^1O_2]_{ss}$ and $[^3C^*]_{ss}$ generally showed positive correlations with $SUVA_{254}$ and $SUVA_{365}$. These correlations provided evidence that the

production of $^1O_2$ and $^3C^*$ was enhanced by aromatic compounds. This enhancement likely occurred though a combination of enhanced rates of light absorption and photosensitization of water-soluble BrC chromophores (Manfrin et al., 2019; Chen et al., 2021). The linear correlations of $[^3C^*]_{ss}$ with $SUVA_{254}$ and $SUVA_{365}$ were noticeably weaker compared to $[^1O_2]_{ss}$. The weaker $[^3C^*]_{ss}$ correlations could be attributed to the sample-to-sample variability in the size of the population of $^3C^*$ species that were able to oxidize syringol.

It is important to note that even though our results (Figures 4b, 4d, and S14) indicated that aromatic compounds were likely key water-soluble BrC constituents and photosensitizers that enhanced $^1O_2$ and $^3C^*$ production, there are other water-soluble BrC constituents and photosensitizers that can also promote $^1O_2$ and $^3C^*$ production. One such example are imidazoles, which are formed from aqueous reactions of dicarbonyls with reduced nitrogen-containing compounds such as amines, ammonium ions, and amino acids (Haan et al., 2009; De Haan et al., 2009, 2011; Kampf et al., 2012; Powelson et al., 2014). Recent studies

have shown that imidazoles can also be formed from aqueous $^3C^*$-photosensitized reactions of phenolic compounds in the presence of ammonium ions (Mabato et al., 2022, 2023). To the best of our knowledge, there has not been a study that have investigated the concentrations of imidazoles in atmospheric PM in Hong Kong. However, imidazoles have been detected in atmospheric PM in urban Guangzhou (another city in South China) (Lian et al., 2022) and at a background forest site in the Nanling Mountains of South China (He et al., 2022). Thus, future studies can focus on identifying other water-soluble BrC

constituents and photosensitizers (e.g., imidazoles) in atmospheric PM in Hong Kong that can play potentially important roles in enhancing $^1O_2$ and $^3C^*$ production.

## 4   Conclusions and implications

In this study, we reported the steady-state concentrations and quantum yields of $^3C^*$ and $^1O_2$ produced by $PM_{2.5}$ in Hong Kong, South China. We quantified the production of $^3C^*$ and $^1O_2$ in illuminated aqueous extracts of $PM_{2.5}$ collected in different

seasons at two urban sites and one coastal semi-rural site during a year-round study. Variations in the WSOC concentrations and light absorption properties of water-soluble BrC across the three sites were found to be statistically insignificant. In contrast, variations in the WSOC concentrations and light absorption properties of water-soluble BrC across the four seasons were significant. Higher concentrations of WSOC and more light absorbing water-soluble BrC were present in the the $PM_{2.5}$ during the fall and winter months. This could be attributed to monsoon-influenced seasonal variations in long-range air mass transport

to Hong Kong. Air masses originating mainly from polluted continental areas located north of Hong Kong contributed to the higher concentrations of WSOC and more light absorbing water-soluble BrC in the the fall and winter $PM_{2.5}$, whereas air masses originating mainly from clean marine regions located south of Hong Kong were responsible for the lower concentrations of WSOC and less light absorbing water-soluble BrC in the summer $PM_{2.5}$.





$^1O_2$ and $^3C^*$ were produced in all the illuminated aqueous extracts of $PM_{2.5}$. The $[^1O_2]_{ss}$ spanned two orders of magnitude,

ranging from $1.56 \times 10^{-14}$ to $1.35 \times 10^{-12}$ M, with a study average of $(4.02 \pm 3.52) \times 10^{-13}$ M. The $[^3C^*]_{ss}$ spanned

two orders of magnitude, ranging from $2.93 \times 10^{-16}$ to $8.08 \times 10^{-14}$ M, with a study average of $(1.09 \pm 1.39) \times 10^{-14}$

M. These $[^1O_2]_{ss}$ and $[^3C^*]_{ss}$ values were in line with the steady-state concentrations previously reported for PM extracts,

fog water, and rain water (Table S7). The $[^1O_2]_{ss}$ and $[^3C^*]_{ss}$ correlated with the concentration of WSOC and the absorbance

of water-soluble BrC, which indicated that water-soluble BrC was a key source of $^1O_2$ and $^3C^*$. Positive linear correlations

between their steady-state concentrations and indicators of aromaticity ($SUVA_{254}$ and $SUVA_{365}$) implied that the production

of $^1O_2$ and $^3C^*$ was enhanced by aromatic compounds, likely though a combination of enhanced rates of light absorption and

photosensitization of water-soluble BrC chromophores. Location (i.e., urban vs. semi-rural) did not have a significant effect

on $[^1O_2]_{ss}$ and $[^3C^*]_{ss}$, which indicated that BrC from local $PM_{2.5}$ sources were likely not the primary drivers of year-round

$^3C^*$ and $^1O_2$ production. In contrast, seasonality had a significant effect on $[^1O_2]_{ss}$ and $[^3C^*]_{ss}$, with higher $[^1O_2]_{ss}$ and $[^3C^*]_{ss}$

observed in the fall and winter compared to the summer. This indicated that the seasonal trends of $^1O_2$ and $^3C^*$ production

in $PM_{2.5}$ in Hong Kong were governed by the seasonal variations in long-range air mass transport. Consequently, regional

$PM_{2.5}$ sources located in continental areas north of Hong Kong contributed to the higher $^1O_2$ and $^3C^*$ production in the fall

and winter.

Even though the steady-state concentrations of ·OH ($[·OH]_{ss}$) were not measured in this study due to insufficient extract

volumes, previous studies have reported that they are typically on the order of $10^{-17}$ to $10^{-15}$ M (Arakaki and Faust, 1998;

Arakaki et al., 1999, 2006, 2013; Anastasio and McGregor, 2001; Anastasio and Jordan, 2004; Anastasio and Newberg, 2007;

Kaur and Anastasio, 2017; Kaur et al., 2019; Manfrin et al., 2019). We hypothesize that the $[·OH]_{ss}$ in our illuminated extracts

are also on the order of $10^{-17}$ to $10^{-15}$ M. The main precursors of ·OH in Hong Kong are likely BrC and inorganic nitrate,

both of which have the highest concentrations in the winter and the lowest concentrations in the summer (Table S2). Therefore,

it is likely that ·OH production will have a similar seasonal trend as $^3C^*$ and $^1O_2$ production. Consequently, the concentrations

of $^3C^*$ and $^1O_2$ can potentially be up to $10^3$ and $10^5$ higher than than the concentrations of ·OH in the extracts, respectively.

Based on work by Kaur et al. (2019) and Ma et al. (2023a), the differences between the $^3C^*$ and $^1O_2$ concentrations vs.

·OH concentrations are expected to be even larger under aerosol liquid water conditions. Thus, despite the lower reactivities

of organic aerosol compounds with $^1O_2$ and $^3C^*$ compared to their corresponding reactvities with ·OH, $^1O_2$ and $^3C^*$ will

likely be present at high enough concentrations that they can be competitive photooxidants to ·OH under aerosol liquid water

conditions (Kaur et al., 2019; Manfrin et al., 2019). This necessitates the inclusion of aqueous reactions involving $^1O_2$ and

$^3C^*$ with organic aerosol compounds into atmospheric models since $^1O_2$ and $^3C^*$ can potentially play important roles in the

photochemical processing of organic aerosol compounds in atmospheric phases due to the high concentrations of $^1O_2$ and $^3C^*$

offsetting their lower reactivities.

The significance of our results lies foremost in the seasonal trends observed for $[^1O_2]_{ss}$ and $[^3C^*]_{ss}$, and how they correlated

with the seasonal variations in the long-range air mass transport. Since many South China cities share similar monsoon-

influenced seasonal air quality and aerosol pollution characteristics as Hong Kong, we anticipate that many South China cities

will have similar seasonal trends of $^1O_2$ and $^3C^*$ production in atmospheric aerosols. In addition, given that their high concen-



trations will likely offset their lower reactivities, $^1O_2$ and $^3C^*$ seasonality in atmospheric aerosols can potentially influence the
aqueous photochemical processing of organic aerosol compounds in South China, a region in which aqueous aerosol chemistry
plays important roles in the formation and transformation of SOA (Li et al., 2013b, a). It should be noted that although our
results showed that the location (i.e., urban vs. semi-rural) did not have a significant effect on $^1O_2$ and $^3C^*$ production in $PM_{2.5}$
in Hong Kong, this may not necessarily be the case for other South China cities, especially those that are located close to areas
with biomass burning activities (Yuan et al., 2015).

While this study reports the first measurements of the quantum yields and steady-state concentrations of $^3C^*$ and $^1O_2$ produced in atmospheric aerosols in South China, there are a number of caveats that should be noted. First, the $[^1O_2]_{ss}$ and $[^3C^*]_{ss}$
values reported in our study serve as lower limits since they were measured using extracts comprised of only the water-soluble
fraction of $PM_{2.5}$. Water-insoluble BrC, which reportedly dominates the total BrC absorption in some parts of China (Bai
et al., 2020; Huang et al., 2020; Wang et al., 2022), will likely produce $^1O_2$ and $^3C^*$ as well. Second, due to limited extract
volumes for photochemical experiments and chemical analysis, only one $^3C^*$ chemical probe was used in our study to quantify $^3C^*$ quantum yields, formation rates, and steady-state concentrations. Hence, we only report concentrations of a subset
of $^3C^*$ species. Measurements of $^3C^*$ quantum yields and steady-state concentrations can be better constrained with the use
of multiple $^3C^*$ probes (Kaur and Anastasio, 2018; Kaur et al., 2019; Ma et al., 2023a, b). Third, photochemical experiments
were performed using diluted extracts. These experimental conditions were substantially more diluted than atmospheric $PM_{2.5}$
conditions. Thus, the concentrations of BrC chromophores in our extracts were substantially lower than those in atmospheric
$PM_{2.5}$, which would influence the reaction kinetics, and consequently $^3C^*$ and $^1O_2$ production. Based on work by Kaur et al.
(2019) and Ma et al. (2023a), higher $[^1O_2]_{ss}$ and $[^3C^*]_{ss}$ in atmospheric $PM_{2.5}$ are expected due to the higher concentrations
of BrC chromophores, though extrapolation from dilute extract conditions to concentrated $PM_{2.5}$ conditions is complex and
non-linear. Fourth, our extracts were not buffered and their average pH was $4.68 \pm 0.29$, whereas the pH of atmospheric $PM_{2.5}$
in Hong Kong has been reported to be between 1.8 and 5.1 (Nah and Lam, 2022; Nah et al., 2023). pH can influence the
composition of protonated vs. unprotonated BrC chromophores, which in turn will affect their absorption and reaction kinetics
(Ma et al., 2021). Fifth, this work focuses on $^1O_2$ and $^3C^*$ production in $PM_{2.5}$ extracts. Previous work on $^1O_2$ production in
illuminated extracts of size-fractionated supermicron-sized road dust (< 45 to 500 µm) suggest that aerosol size may influence
$^1O_2$ production (Cote et al., 2018). At present, it is unclear how aerosol size within atmospheric $PM_{2.5}$ influences $^1O_2$ and
$^3C^*$ production. Hence, the effects of dilution, pH, and aerosol size on photooxidant production from both water-soluble and
water-insoluble BrC in atmospheric PM should be explored in future studies to further our understanding of aqueous organic
aerosol photochemistry in the South China region.

*Data availability.* Light absorption and kinetic data have been submitted to the data repository Zenodo (https://doi.org/10.5281/zenodo.7827983).
Data can also be made available upon request to the corresponding author (theodora.nah@cityu.edu.hk).



545 *Author contributions.* YLyu and TN designed the study. YHL collected the field samples. YLyu performed the chemical analysis and experiments. YLyu, YLi, NBD, and TN analyzed the data. YLyu and TN prepared the manuscript with contributions from all co-authors.

*Competing interests.* One of the authors is a member of the editorial board of *Atmospheric Chemistry and Physics*. The peer-review process was guided by an independent editor, and the authors also have no other competing interests to declare.

*Acknowledgements.* We acknowledge the assistance of Yanhao Miao with the backward trajectory analysis, Jason Lam for his assistance
550 with the furfuryl alcohol distillation, Wing Chi Au and Chung Ming Tai for their help on filter extraction and/or chemical analysis. The work described in this paper was supported by a grant from the Research Grants Council of the Hong Kong Special Administrative Region, China (Project No. 11303720).



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
