# Peer review of "Seasonal variations in the production of singlet oxygen and organic triplet excited states in aqueous PM2.5 in Hong Kong, South China"

_EGUsphere, 2023_

## Author Comment (AC1)

We thank the referees for their careful reading and the detailed comments. The responses to the comments of the two referees in our direct reply (shown below) and within the revised manuscript (see marked copy) are provided below. The pages and lines indicated below correspond to those in the marked copy.

**Response to Referee 1 (Referee's comments are italicized)**

1. Referee comment: "*The methods section is well documented, and the experiments well described. I have one small technical comment about the determination of the rate of light absorption (equation 2): In equation 2, it is not clear what the authors used for the optical pathlength, is it the diameter of the quartz tubes or some average path length through the tubes? A side question on that point is if an incorrect pathlength in the solution would influence the singlet oxygen quantum yield determination. I am wondering about the presented singlet oxygen quantum yield numbers that are higher than in previous studies. The Rayonet reactor used by the authors has reflecting side walls and the effective path length could be longer than the measured one due to photon passing multiple time through the experiment's solution.*"

**Author response:** We assumed that the path length of the light is equal to the inner diameter of the quartz tubes (1.25 cm). We acknowledge that the actual optical path length may be slightly higher than the inner diameter of the quartz tubes used in our calculations. However, based on previous work by Ossola et al. (2021), we do not expect these differences to affect our $R_{abs}$ and quantum yield values significantly. To confirm this, we performed calculations to determine the extent to which the quantum yields change when a smaller optical path length (1 cm) was used to calculate $R_{abs}$. We found that the quantum yield values for both $^1O_2^*$ and $^3C*$ were 0.06 to 1.56 % (0.53 % on average) lower relative to when the optical path length was set to 1.25 cm. This information has been added into the revised manuscript.

We thank the referee for providing some suggestions as to why our reported $^1O_2^*$ quantum yields are higher than previously reported values. However, we do not think that our use of the Rayonet photoreactor in photochemical experiments contributed to our reported $^1O_2^*$ quantum yields being higher than previously reported values. The reflecting side walls in Rayonet photoreactor likely did not prolong the optical path length too much since all the photochemical experiments were performed in batch. The quartz tubes filled with different solutions were placed cylindrically on the carousel sample plate during each photochemical experiment, which would reduce the reflection from the side walls. In addition, as we have discussed above, the optical path length does not significantly affect the calculated quantum yields. Overall, we believe that our reported high $^1O_2^*$ quantum yields are due primarily to the composition and age of the water-soluble BrC in the $PM_{2.5}$ investigated in our study, rather than being due to inaccuracies in the optical path length originating from our use of the Rayonet photoreactor in photochemical experiments or from our assumption that the path length of the light is equals to the inner diameter of the quartz tubes.

The following changes have been made to the revised manuscript:

**Page 6, line 136: "where $d$ is the path length of the light through the quartz tubes used in the photochemical experiments (cm), $10^3$ is for units conversion (cm$^3$ L$^{-1}$), $I_{0,\lambda}$ (mol-photons nm$^{-1}$ s$^{-1}$ cm$^{-2}$) is the absolute irradiance of the light source at wavelength λ, and Δλ is the interval of wavelength (1 nm). $d$ was assumed to be equals to the inner diameter of the quartz tubes (1.25 cm). However, we acknowledge that the actual optical path length may be slightly different from the inner diameter of the quartz tubes used in our**

**calculations. Nevertheless, we do not expect these differences to affect our $R_{abs}$ and quantum yield calculations significantly (Ossola et al., 2021). For instance, using $d = 1$ cm will cause the calculated quantum yields to decrease, on average, only by 0.53 % relative to quantum yields calculated using $d = 1.25$ cm."**

2. Referee comment: "*Reading the results and discussion, I was left a little wondering about the reasons for the observed seasonality. Maybe the authors could elaborate a little more. Here are some of my thoughts on the subject:*

*The authors did not see significant differences in the extracts between the three sampling sites (and attributed that to the brown carbon source being mostly not local) but observed a seasonal difference in extracts characteristics. Reading the article, I understand that the authors attributed the winter brown carbon to mainland China sources. I was left wondering about what the summer brown carbon sources are. Are the authors attributing the summer provenance to local sources, marine emissions or on lands further apart from Hong Kong? It would be worth being clearer about this point.*

*This could have some implications if the summer aerosols are older than the winter ones and could explain some of the seasonal differences. Literature on photobleaching indicate that light exposure induces a loss of sensitizing properties (Water Research Volume 66, 1 December 2014, Pages 140-148 Photobleaching-induced changes in photosensitizing properties of dissolved organic matter) and a loss in absorbance (Environ. Sci. Technol. 2021, 55, 13152−13163). The authors observed an increased singlet oxygen quantum yield for the fall and winter extracts. If the summer extracts were older and more exposed to sunlight that could explain part of the observed seasonal difference.*

*A last thought on the high singlet oxygen quantum yield observed is that ozone exposure may induce an increase in singlet oxygen quantum yield (Environ. Sci. Technol. 2019, 53, 5622−5632). If the authors think that their extracts were more exposed to ozone than extracts from other (north American and European studies), that could be a possible explanation.*"

**Author response:** Hong Kong's air quality is influenced by the East Asian monsoon system. In the fall and winter months, northerly prevailing winds carry dry polluted air masses from northern continental regions to Hong Kong. The referee is correct in stating that the main contributors to $PM_{2.5}$ and water-soluble BrC in fall and winter are polluted continental areas located north of Hong Kong (e.g., Mainland China). In the summer, southerly prevailing winds carry mostly clean marine air masses from the southern sea areas to Hong Kong. Since these clean marine summer air masses typically have low levels of $PM_{2.5}$ and water-soluble BrC, thus local sources (e.g., vehicle emissions, combustion related activities, and solvent usage) are the main contributors to $PM_{2.5}$ and water-soluble BrC in the summer. This has been shown in several previous studies. We have revised the manuscript to make these points clearer.

We thank the referee for providing other possible suggestions to explain the observed seasonality in the steady-state concentrations of $^1O_2{}^*$ and $^3C^*$. It is true that photobleaching can induce a loss of photosensitization properties of water-soluble BrC, and consequently affect the quantum yields for $^1O_2{}^*$ and $^3C^*$. However, it is important to note that the seasonal variations in the $^1O_2{}^*$ and $^3C^*$ quantum yields are not statistically significant (Figure 4). In fact, the winter quantum yield values are quite close to the summer quantum values even though photobleaching is supposedly most extensive in the summer due to the high solar irradiance. Nevertheless, we cannot discount the possibility that the summer BrC chromophores may have

been more effective in producing photooxidants but the enhanced photobleaching caused by the stronger solar irradiation in the summer led to weakened photosensitization ability and consequently similar quantum yields across the seasons. We have added this possibility into our revised manuscript.

We agree that it is possible that exposure to ambient ozone pollution may have contributed to our reported $^1O_2^*$ quantum yields being higher than previously reported values. We have added this possibility into our revised manuscript.

The following changes have been made to the revised manuscript:

**Page 10, line 256: "Air masses originating mainly from polluted continental areas located north of Hong Kong contributed to the high PM$_{2.5}$ and WSOC concentrations in fall and winter (Figures S6 to S8). In the summer, air masses originate from clean marine regions located south of Hong Kong instead. These summer marine air masses generally have low PM$_{2.5}$ and WSOC concentrations. This results in Hong Kong having substantially lower PM$_{2.5}$ and WSOC concentrations in the summer compared to the fall and winter. Consequently, regional sources are the main PM$_{2.5}$ contributors in fall and winter, whereas local sources are the main PM$_{2.5}$ contributors in the summer (Huang et al., 2014; Zhang et al., 2018; Chow et al., 2022)."**

**Page 11, line 287: "It is possible that the observed higher absorbance and aromaticity in the urban CU and TW extracts were due to the presence of oxygenated aromatic compounds (e.g., highly substituted phenolic compounds) from local anthropogenic sources such as vehicle emissions, combustion-related (e.g., cooking, power generation and usage) activities, and solvent usage (Guo et al., 2003; Chen et al., 2017; Cui et al., 2018; Bilal et al., 2019)."**

**Page 19, line 463: "The variations in $\Phi_{1O2*}$ and $\Phi_{3C*}$ across the four seasons were not statistically significant ($p > 0.05$), which indicated that seasonality did not have a significant effect on the photosensitization efficiencies of $^1O_2^*$ and $^3C^*$. However, we cannot discount the possibility that the statistically insignificant variations in $\Phi_{1O2*}$ and $\Phi_{3C*}$ across the four seasons could be due to photobleaching. Leresche et al. (2021) previously reported reduced $^1O_2^*$ photosensitization for the extracts of summer PM$_{2.5}$ collected from Colorado, USA due to enhanced photobleaching. Thus, it is possible that the summer BrC chromophores may have been more effective in producing photooxidants but the enhanced photobleaching caused by stronger solar irradiation led to their weakened photosensitization ability and consequently resulted in statistically insignificant variations in $\Phi_{1O2*}$ and $\Phi_{3C*}$ across the seasons in our study."**

**Page 14, line 348: "For instance, ozone is a major ground-level air pollutant in Hong Kong (Liao et al., 2021). Exposure to ambient ozone pollution could have led to higher $\Phi_{1O2*}$ values due to the formation of quinone-like moieties from ozone aging of phenolic moieties present in water-soluble BrC (Leresche et al., 2019)."**

3. Referee comment: "*Line 254 and Lines 268-269: 'due to the presence of aromatic compounds (e.g., polycyclic aromatic hydrocarbons) from local vehicle emissions' and 'These results implied that the water-soluble BrC in PM$_{2.5}$ was weakly influenced by local emission sources near the sites.' These two sentences look to be saying first that local emission sources are important and second that it is not important.*"

**Author response:** The first sentence was intended to explain the higher average SUVA values for the urban CU and TW sites compared to the semi-rural HT site. However, the ANOVA tests subsequently performed for the WSOC concentrations and optical properties indicated that the differences were actually not statistically significant. The second sentence was meant to explain why the WSOC concentrations and optical properties for the urban CU and TW sites vs. semi-rural HT site were statistically insignificant. We acknowledge that our wording in the second sentence was a little confusing. Thus, we have made the following changes in the revised manuscript:

**Page 12, line 308: "These results indicated that the locations (i.e., urban vs. semi-rural) did not have a significant influence on the concentration of WSOC and light absorption properties of water-soluble BrC in PM$_{2.5}$."**

4. Referee comment: "*Line 332: 'On average, $R_{abs}$ was about 20 times higher than the sum of $Rf,^1O_2$ and $Rf,^3C*$. This indicated that majority of the (photo) energy absorbed by the illuminated extracts in the photochemical experiments were dissipated by non-reactive pathways' this paragraph is misleading. $Rf,^3C*$ is a very small subset of the total triplets. The total triplets rate of light absorption can be estimated to be around 3 times $Rf,^1O_2$. The factor 3 coming from the estimate of the yield of triplet state conversion to singlet oxygen found in Environ. Sci. Technol. 2017, 51, 13151−13160. Also, the authors should not sum $Rf,^1O_2$ and $Rf,^3C*$ as singlet oxygen if formed from the triplet states.*"

**Author response:** We agree. We have removed the above-mentioned discussion from the revised manuscript.

**Response to Referee 2 (Referee's comments are italicized)**

1. Referee comment: "*One of the difficulties with reporting $^3C*$ and $^1O_2$ concentrations in particle extracts is that the results depend on the extract concentration, i.e., the PM mass/liquid water mass ratio. In relatively dilute extracts, oxidant concentrations are proportional to the extract concentration, so that changes in dilution lead to significant changes in $[^3C*]$ and $[^1O_2]$. This complicates comparing oxidant concentrations, both across and within studies, as they will vary with the amount of water used for extract preparation as well as the ambient PM mass concentration. (Fortunately, the authors used a constant sampling flow rate and sampling time.) Thus there's not much meaning to statements such as "The range of $[^1O_2]_{ss}$ values is remarkably large…" (Line 288). From an environmental perspective, two aerosols with the same PM composition but very different PM mass concentrations (i.e., µg/m$^3$) would have roughly the same concentration of $^3C*$ and $^1O_2$ in their particle water. However, concentrations of the two oxidants in the PM extracts (assuming constant sampling time and solvent volume) would be very different.*

*It would be helpful to discuss this issue at the beginning of the results section. As part of this, the authors should report their PM mass/water mass ratios (more about this below) and explain where their extracts fall on the rain – fog/cloud – aerosol liquid water (ALW) continuum. It would also be helpful to explain how much of the concentration variation that they report is due to differences in airborne PM mass concentrations or collected PM masses.*

*It would also be helpful to report DOC-normalized production rates, i.e., $Rf(^1O_2)/WSOC$ and $Rf(^3C^*)/WSOC$, and how they compare to past work. These are important parameters for estimating oxidant concentrations in ALW.*"

**Author response:** Sentences such as "The range of $[^1O_2^*]_{ss}$ values is remarkably large…" have been removed in the revised manuscript. As requested, we have added a discussion on the calculation of $PM_{2.5}$ mass/water mass ratios of the extracts in the revised manuscript. It should be noted that in addition to keeping the sampling flow rates and periods consistent, we were also consistent in all of our filter extraction protocols. Based on the volume of Milli-Q water used for filter extraction followed by additional dilution during the preparation of every extract used for measurements and photochemical experiments, we determined that our protocols were equivalent to each filter being extracted in 15.54 mL Milli-Q water. This allowed us to compare the WSOC concentrations, light absorption properties, and photooxidants across the extracts. Our calculations showed that the $PM_{2.5}$ mass/water mass ratios of the extracts ranged from $1.86 \times 10^{-5}$ to $2.14 \times 10^{-4}$ µg PM/µg $H_2O$, which are close to cloud/fog water conditions but are much more diluted than aerosol liquid water conditions. Due to the non-linear relationship of oxidant concentrations on the $PM_{2.5}$ mass/water mass ratios and limited filter samples, we did not perform additional experiments (e.g., those performed by Kaur et al. (2019 and Ma et al. (2023a)) to predict oxidant concentrations in ambient aerosol liquid water based on our measurements. We observed a good linear correlation between the WSOC concentration and $PM_{2.5}$ mass/water mass ratio ($r^2 = 0.93$). This suggested that the seasonal variations in the WSOC concentration could be attributed to the seasonal variations in the $PM_{2.5}$ mass concentration. Further statistical analyses performed revealed that seasonal variations in the steady-state concentrations of $^1O_2^*$ were likely driven primarily by the $PM_{2.5}$ mass concentration and WSOC concentration (please refer to our response to the referee's comment 19). In addition, we have added the requested comparisons of $R_{f,1O2*}/[WSOC]$ and $R_{f,3C*}/[WSOC]$ across studies into the revised manuscript.

The following changes have been made to the revised manuscript:

**Page 5, line 114: "Extracts from three consecutive sampling periods (9 filters in 9 days) were aggregated to minimize daily variability. This procedure resulted in roughly 3 aggregated extracts per season for each site, referred to by the site and sampling start date. For example, sample CU041220 refers to extracts of filters collected from 4 Dec 2020 to 13 Dec 2020 at the CU site. Due to sampler pump malfunction, filters were not collected at the CU site from 18 June 2020 to 24 June 2020 and at the HT site from 18 April 2020 to 27 April 2020. In addition, some aggregated extracts were comprised only of two consecutive sampling periods (6 filters in 6 days) due to limited filter samples. It should be noted that all the aggregated extracts were further diluted with Milli-Q water by a factor of 2.22 for light absorption measurements and photochemical experiments. This was equivalent to extracting each filter with 15.54 mL Milli-Q water. The $PM_{2.5}$ mass to water mass ratios ($PM_{2.5}$ mass/$H_2O$ mass) were calculated for each aggregated extract using the ambient $PM_{2.5}$ mass concentrations measured at or near the sampling sites by the Hong Kong Environmental Protection Department. Detailed information about the sampling periods, allocation of aggregated extracts, and calculation of $PM_{2.5}$ mass/$H_2O$ mass values are shown in Table S1."**

**Table S1.** List of aggregated extracts for CU, TW, and HT.

| Season | CityU | Tsuen Wan | Hok Tsui |
|---|---|---|---|

| | Sample ID | Total sets[a] (72 h/set) | Mass ratio[b] ($10^{-4}$) | Sample ID | Total sets[a] (72 h/set) | Mass ratio[b] ($10^{-4}$) | Sample ID | Total sets[a] (72 h/set) | Mass ratio[b] ($10^{-4}$) |
|---|---|---|---|---|---|---|---|---|---|
| Winter | CU041220 | 3 | 2.11 | TW110221 | 3 | 1.43 | HT051221 | 3 | 1.67 |
| | CU131220 | 3 | 1.36 | TW200221 | 2 | 1.46 | HT140121 | 3 | 2.01 |
| | CU221220 | 2 | 1.68 | TW260221 | 2 | 1.16 | HT230121 | 2 | 1.47 |
| Spring | CU110321 | 3 | 1.33 | TW190521 | 3 | 0.49 | HT090421 | 3 | 0.95 |
| | CU200321 | 3 | 1.58 | TW280521 | 3 | 0.71 | HT270421 | 2 | 0.84 |
| | CU290321 | 3 | 1.01 | TW060621 | 3 | 0.91 | N.A. | | |
| Summer | CU240621 | 3 | 0.82 | TW160721 | 3 | 0.86 | HT130821 | 3 | 0.22 |
| | CU030721 | 3 | 0.58 | TW250721 | 3 | 1.08 | HT220821 | 3 | 0.19 |
| | N.A. | | | TW030821 | 3 | 1.12 | HT310821 | 3 | 1.87 |
| Fall | CU100921 | 2 | 0.87 | TW161121 | 3 | 1.71 | HT181021 | 3 | 0.62 |
| | CU160921 | 2 | 0.98 | TW251121 | 3 | 1.24 | HT271021 | 3 | 1.14 |
| | CU250921 | 3 | 1.48 | TW061221 | 3 | 2.14 | HT051121 | 3 | 0.69 |

**Note: Due to sampler pump malfunction, filters were not collected at the CU site from 18 June 2020 to 24 June 2020 and at the HT site from 18 April 2020 to 27 April 2020.**

**a. Each sample set was collected continuously for 72 hours. For sample IDs that were comprised of three sets of filters (e.g., CU041220), this meant that the aggregated extracts were comprised of three consecutive 72-h sampling periods (9 days in total). For sample IDs that were comprised of two sets of filters (e.g., CU100921), this meant that the aggregated extracts were comprised of two consecutive 72-h sampling periods (6 days in total).**

**b. The $PM_{2.5}$ mass/water mass ratio (µg $PM_{2.5}$/µg $H_2O$) was calculated by taking the ratio of the $PM_{2.5}$ mass divided by the water mass for each aggregated extract sample. The $PM_{2.5}$ mass was calculated using the daily $PM_{2.5}$ mass concentration measured at or near the sampling sites by Hong Kong Environmental Protection Department (HKEPD) (https://cd.epic.epd.gov.hk/EPICDI/air/station/?lang=en). Since the CityU sampling site did not have a $PM_{2.5}$ mass monitor, the $PM_{2.5}$ mass concentration data at the closest HKEPD monitor site (Sham Shui Po, 1.5 km from CityU) was used to calculate the mass ratio for CityU samples. The $PM_{2.5}$ mass concentration data for Hok Tsui was not publicly available, and had to be requested from the HKEPD. Since a consistent extraction protocol and constant dilution ratio were applied to each aggregated sample, the $PM_{2.5}$ mass to water mass ratios were calculated on a per filter basis. To obtain the $PM_{2.5}$ mass collected onto each filter, the 9-day or 6-day averaged $PM_{2.5}$ mass concentration was multiplied by the filter sampler's flow rate (we used 29 L/min in our calculations since the sampling flow rate decreased from of 30 L $min^{-1}$ to 28 L $min^{-1}$ over the 72-h continuous sampling period) and sampling time (72-h × 60 min). The mass ratios were calculated under the same conditions as in photochemical experiments (i.e., measurement of $^1O_2^*$ and $^3C^*$), which was equivalent to extracting each filter in 15.54 mL Milli-Q water. These values served as an upper bound due to materials lost during water extraction and filtration process.**

**Page 9, line 241:** "The same sampling flow rate (30 L min$^{-1}$) and period (72 h) were used to collect all the filters and the same dilution ratio (i.e., equivalent to extracting each filter in 15.54 mL Milli-Q water) was used to prepare all the extracts. This allowed us to compare the WSOC concentrations and light absorption properties across the extracts. The PM$_{2.5}$ mass/water mass ratios for the extracts (Table S1) ranged from $1.86 \times 10^{-5}$ to $2.14 \times 10^{-4}$ µg PM$_{2.5}$/µg H$_2$O, which were close to fog and cloud water conditions but were much more diluted compared to aerosol liquid water conditions (ca. 1 µg PM/µg H$_2$O) (Liao and Seinfeld, 2005; Herrmann et al., 2015; Nguyen et al., 2016; Seinfeld and Pandis, 2016). The concentrations of WSOC in the extracts ranged from 3.8 to 25.7 mg-C L$^{-1}$, with a study average of 13.7 mg-C L$^{-1}$ (Table S2), which were close to the WSOC concentrations previously measured in fog and ground base clouds (Herckes et al., 2013). The concentrations of WSOC in the extracts were linearly correlated (SLR $r^2$ = 0.93) with the PM$_{2.5}$ mass/water mass ratios (Figure 2)."

[Figure]

**Figure 2. The WSOC concentration as a function of the PM$_{2.5}$ mass/water mass ratio for the extracts. Blue, green, red, and orange symbols denote the winter, spring, summer, and fall samples, respectively. The dashed lines represent 95 % prediction bands. The SLR $r^2$ and Pearson's $r$ are the coefficient of determination for simple linear regression and the Pearson correlation coefficient, respectively.**

**Page 14, Line 338:** "The study average WSOC-normalized R$_{f,1O2*}$ (($6.95 \pm 4.28) \times 10^{-9}$ M s$^{-1}$ L mg-C$^{-1}$) was within a factor of 2 of previously reported values for PM$_{2.5}$ samples collected in urban and rural areas in Colorado, USA (Leresche et al., 2021) and for PM samples collected in biomass burning-influenced areas in California, USA (Kaur et al., 2019; Ma et al., 2023a)."

**Page 16, Line 385:** "The study average WSOC-normalized R$_{f,3C*}$ (($6.51 \pm 7.90) \times 10^{-10}$ M s$^{-1}$ L mg-C$^{-1}$) was 3 to 7 times lower than the previously reported value for PM samples collected in biomass burning-influenced areas in California, USA (Kaur et al., 2019; Ma et al., 2023a)."

2. Referee comment: "*The meaning of BrC "quality" is unclear. On Line 425 it appears to be defined as "specific absorbance" (Line 425), but this term is a bit vague. It also seems that any definition of BrC quality should include the efficiency of oxidant formation, i.e., quantum yield.*"

**Author response:** To remove any ambiguity, we have made the following changes in the revised manuscript:

**Page 2, line 42: "The production of $^3C*$ and $^1O_2^*$ are influenced by both the concentrations (i.e., quantity) and quantum yields (i.e., quality) of BrC chromophores (Bogler et al., 2022). The quantum yield, which describes the efficiency of oxidant photosensitization, can be obtained from dividing the number of moles of oxidant generated by the number of moles of photons absorbed by the photosensitizer."**

**Page 20, line 498: "The quantity of the BrC chromophores is associated with their concentrations, whereas the quality is associated with their quantum yields and WSOC-normalized light absorption properties (e.g., MAC and SUVA values). In other words, some BrC chromophores are more efficient as making photooxidants, and thus PM$_{2.5}$ with higher quantum yields can be considered to have higher quality BrC chromophores towards $^1O_2^*$ and $^3C^*$ formation.**

3. Referee comment: "*Section 2.1.2 (Sampling and extraction protocols) needs more details, in part so there's a record of sample dilution to aid with estimating oxidants under ALW conditions in the future. For example, how much Milli-Q water was used to extract a filter? When the consecutive filters were combined to make a sample, how much additional water was added to reach "an adequate volume"? If the dilution parameters were variable, the information should be put in Table S1. It would also be helpful to include the PM mass/water mass ratio of each extract. Can this be estimated based on what was measured and/or from nearby ambient PM$_{2.5}$ monitors? Two other experimental methods questions that should be addressed: How long were filters vortexed? How long were extracts stored in the refrigerator before illumination?*"

**Author response:** We refer the referee to our response to his comment 1 regarding the addition of a description of our protocols for sample dilution and the calculation of PM mass/water mass ratio for each sample to the revised manuscript. In addition, we have added the requested information regarding our sampling and extraction protocols to the revised manuscript:

**Page 5, line 106: "Each filter was extracted in 7 mL Milli-Q water inside a 15 mL sterile centrifuge tube (JET BIOFIL®) by vortexing for 4 minutes (MX-S DLAB, medium high power). The disintegrated filter parts were removed from the extracts by filtration using 0.22 μm pore size nylon syringe filters (Nylon66, Jinteng®). The filtered extracts were stored in amber vials at 4 °C in a refrigerator until the day of photochemical experiments. The maximum amount of time for which the extracts were stored in the refrigerator (i.e., from the day of extraction to the day of project completion) is 6 months. We compared the WSOC and light absorption measurements performed on the extracts within a week of extraction vs. after all the photochemical experiments have concluded, and observed minimal changes in the WSOC and light absorption properties of the extracts."**

4. Referee comment: "*Section 2.6. The disadvantage of SYR (and TMP) as a probe is that its decay can be inhibited by DOM and Cu, which leads to an underestimate of the oxidizing triplet*

*concentration, as initially described in surface waters. Inhibition can be very important in PM extracts, especially if highly concentrated. In our 2018 and 2019 work we didn't know this was an issue; the current manuscript seems to be in the same boat. We discuss inhibition, how to correct it, and the original surface water references, in our more recent papers (Ma et al., 2023a and 2023b; also Ma et al., https://doi.org/10.5194/egusphere-2023-861). In this third reference we report SYR inhibition factors for a year of samples: at DOC ~ 25 mg/L, the upper range of the extracts in the Lyu et al. manuscript, we measured inhibition factors (IF) as low as ~ 0.5. While the IF depends both on DOC composition and concentration, as well as Cu concentrations, our result suggests that correcting for inhibition in the current manuscript would increase [3C\*] by up to a factor of two. The authors should add a discussion about inhibition and its potential impact on the current work.*"

**Author response:** As requested, we have expanded on our discussion of the inhibition effects of DOM and copper in the revised manuscript:

**Page 15, line 371: "In addition, the decays of oxidizing $^3C^*$ chemical probes (e.g., syringol and 2,4,6-trimethylphenol) can be inhibited by the co-presence of some atmospheric species (e.g., copper, water-soluble organic matter), especially under highly concentrated conditions (Canonica and Laubscher, 2008; Maizel and Remucal, 2017; Mccabe and Arnold, 2017; Ma et al., 2023b, c). Using the equations provided by Ma et al. (2023b), we estimate that our reported $[^3C^*]_{ss}$ values may be underestimated by as much as a factor of 2 due to water-soluble organic matter inhibiting the decay of syringol. In addition, water-soluble copper, another atmospheric species known to inhibit syringol decay (Ma et al., 2023c), can be present in substantial concentrations in $PM_{2.5}$ in some urban areas in Hong Kong (Yang et al., 2023). However, the extent to which water-soluble copper will impact $[^3C^*]_{ss}$ values is currently unknown."**

5. Referee comment: "*Section 3.4. Earlier in the manuscript, the authors found that oxidant concentrations were strongly (for $^1O_2$) or weakly (for $^3C^*$) correlated with both WSOC and α(300). In section 3.4 they examine oxidant concentrations versus MAC(300) or SUV(254): these correlations are similar or slightly weaker to the cases with WSOC and α(300). The former correlations make more sense, in that both the oxidant concentrations, WSOC, and α(300) all depend on the concentration of the PM extract. In contrast, the latter correlations are examining concentrations, which depend on extract concentrations, with absorbance measures (MAC(300) or SUV(254)) that should be independent of extract concentrations. Given all of this, it seems better to show the WSOC and α(300) correlations in the main text and move the MAC(300) and SUV(254) results to the SI.*"

**Author response:** While we understand the reasons provided by the referee in encouraging us to show the WSOC and $α_{300}$ correlations in the main manuscript and move the $MAC_{300}$ and $SUVA_{254}$ results to the SI, we prefer to keep the figures as they are. First, our main intention for showing the correlations between oxidant ($^1O_2^*$ and $^3C^*$) concentrations and $MAC_{300}$ is to examine more closely how $^1O_2^*$ and $^3C^*$ production depends on water-soluble BrC. We used $MAC_{300}$ because (unlike $α_{300}$) it is a light absorbance parameter that accounts for WSOC dilution. Second, our main intention for showing the correlations between oxidant ($^1O_2^*$ and $^3C^*$) concentrations and $SUVA_{254}$ is to evaluate the contributions of aromatic compounds to $^1O_2^*$ and $^3C^*$ production. As explained in the original manuscript, this it to test our hypothesis that aromatic compounds are important water-soluble BrC species that contributed to $^1O_2^*$ and $^3C^*$ production in our study. $SUVA_{254}$ and $SUVA_{365}$ are commonly used indicators of

aromaticity. Thus, for these reasons, we prefer to show the correlations between oxidant ($^1O_2^*$ and $^3C^*$) concentrations and $MAC_{300}$ and $SUVA_{254}$ in the main manuscript.

6. Referee comment: "*There are a few opportunities to shorten the manuscript. Most significantly, the parameter [Ox]/WSOC is roughly an intermediate step between the previously examined [Ox] and QY(Ox), both in terms of the parameter as well as the results. At this point in the manuscript, [Ox]/WSOC feels repetitious and doesn't offer much that is new: I recommend moving the [Ox]/WSOC results and discussion to the supplement.*

*There are other examples of repetition that should be removed, e.g., (1) the paragraph on lines 424 – 431 repeats ideas that were raised in the original discussion of Figure 3 and (2) comments 18 and 19 under Other Points below.*"

**Author response:** As requested, we have shortened the above-mentioned discussion in the revised manuscript:

**Page 21, line 515: "The important role that the quantity of BrC chromophores plays in driving $^1O_2^*$ and $^3C^*$ production is further emphasized by the weakened seasonal trends of WSOC-normalized $[^1O_2^*]_{ss}$ and $[^3C^*]_{ss}$ (Section S3 and Figure S14)."**

7. Referee comment: "*I also have two suggestions for future work that the authors are free to take or ignore. The first is simple: use simulated sunlight rather than a narrow wavelength band to obtain results that are more directly relevant to atmospheric conditions. The second suggestion is difficult: Strive to measure oxidant concentrations under particle water conditions. In Kaur et al. (2019) and Ma et al. (2023b), we estimated ALW concentrations of photooxidants by extrapolating from three series of dilutions of PM extracts. But even the most concentrated of these extracts are far from ambient conditions, resulting in a large (enormous?) amount of uncertainty in the ALW estimates. How can we as oxidant afficionados use different experimental methods to better determine particle photooxidants, whether it involves probes or other approaches?*"

**Author response:** We appreciate the referee's suggestions for future work. We will keep his suggestions in mind as we plan our future experiments. However, there are always pros and cons with choosing the light source. For instance, while the simulated sunlight provides atmospheric relevant results and direct implication to atmospheric models, it would lead to much longer experimental times, which is a challenge for studies with large number of samples, especially those with low light absorption properties.

8. Referee comment: "*The title starts with "Efficient production", but the quantum yields of triplets are low, indicating inefficient production of this oxidant compared to past samples.*"

**Author response:** We acknowledge that the referee raised a valid point. We have changed the title to "Seasonal variations in the production of singlet oxygen and organic triplet excited states in aqueous $PM_{2.5}$ in Hong Kong, South China".

9. Referee comment: "*A sentence summarizing the quantum yield results in the abstract would be helpful*."

**Author response:** As requested, we have added information about the quantum yields into the abstract of the revised manuscript:

**Page 1, line 15: "The quantum yields of $^1O_2^*$ and $^3C*$ also spanned wide ranges across samples, with a range of 1.19 to 13.74 % and an average of (5.19 ± 2.63) % for $^1O_2^*$, and a range of 0.05 to 3.24 % and an average of (0.56 ± 0.66) % for $^3C*$."**

10. Referee comment: "*Line 123. It's not clear what is meant by "bandwidth". Is it the wavelength range for the lamp output*?"

**Author response:** Yes. We have revised the manuscript as follows:

**Page 6, line 142: "A wavelength range of 290 to 600 nm was used to cover both the output of the photoreactor lamps and light absorption range of all the extracts."**

11. Referee comment: "*Line 254. PAHs are likely a minor contributor to BrC in these water extracts. Is biomass burning, which emits more water-soluble aromatic BrC species, significant in the region? This seems a likely source of aromatic BrC in winter*."

**Author response:** We agree with the referee that due to their generally poor water solubilities, high molecular weight PAHs are likely minor contributors to water-soluble BrC. Instead, we think that it is more likely that water-soluble oxygenated aromatics (e.g., highly substituted phenolic compounds) are major contributors to water-soluble BrC, and we have revised the manuscript accordingly. With regards to the referee's comment about the contribution of biomass burning to winter water-soluble BrC, biomass burning is regarded as a minor local PM$_{2.5}$ source in high urbanized Hong Kong. Thus, we do not expect local biomass burning emissions to contribute to water-soluble BrC in the winter extracts. However, continental areas located north of Hong Kong (especially rural regions in Mainland China) can have high levels of biomass burning activities in the fall and winter seasons. It is likely that air masses from these regions were transported to Hong Kong in the fall and winter seasons. This long-range air mass transport (which is influenced by the East Asia monsoon system) could explain the higher aromaticity in the fall and winter extracts. The following changes have been made to the revised manuscript:

**Page 11, line 285: "These average SUVA$_{254}$ and SUVA$_{365}$ values for the three sites indicated that the organic matter in the urban CU and TW extracts, on average, had higher aromaticity than those in the semi-rural HT extracts (Table 1). It is possible that the observed higher absorbance and aromaticity in the urban CU and TW extracts were due to the presence of oxygenated aromatic compounds (e.g., highly substituted phenolic compounds) from local anthropogenic sources such as vehicle emissions, combustion-related (e.g., cooking, power generation and usage) activities, and solvent usage (Guo et al., 2003; Chen et al., 2017; Cui et al., 2018; Bilal et al., 2019). Upon grouping the SUVA$_{254}$ and SUVA$_{365}$ datasets based on seasonality irrespective of the sampling location, the average seasonal SUVA$_{254}$ and SUVA$_{365}$ values indicated that the organic matter in the fall and winter extracts, on average, had higher aromaticity than those in the spring and summer extracts (Table 2). The higher aromaticity in the fall and winter extracts was**

**likely due to strong biomass burning contributions to ambient fall and winter PM$_{2.5}$. Hong Kong generally has low levels of biomass burning activities. However, fall and winter PM$_{2.5}$ in continental areas north of Hong Kong (e.g., parts of Mainland China) can have substantial contributions from biomass burning, especially in rural areas where residential biomass burning are used for intensive heating purposes (Chen et al., 2017). It is possible that biomass burning-influenced air masses from these northern continental areas were transported to Hong Kong during fall and winter, and consequently contributed to the higher aromaticity in these extracts."**

12. Referee comment: "*Line 263. WSOC and alpha(300) values depend on sample concentration, which will be influenced by the extent of dilution as well as PM mass collected (as described above). So it's not clear that these parameters can be meaningfully compared across samples.*"

**Author response:** The referee is correct in stating that WSOC and $\alpha_{300}$ can be affected by dilution factors and original PM mass collected for each sample. However, we used the same PM$_{2.5}$ sampling flow rate and dilution factor for all the samples, which allowed us to compare these parameters across samples. We recognize the potential for confusion, and thus have made the following changes in the revised manuscript:

**Page 9, line 241: "The same sampling flow rate (30 L min$^{-1}$) and period (72 h) were used to collect all the filters and the same dilution ratio (i.e., equivalent to extracting each filter in 15.54 mL Milli-Q water) was used to prepare all the extracts. This allowed us to compare the WSOC concentrations and light absorption properties across the extracts."**

13. Referee comment: "*Section 2.6. (a) The correction procedure for "direct photolysis" of SYR is mathematically fine, but the description should be corrected: SYR shouldn't undergo direct photodegradation since it does not absorb light in the range of their lamp. The loss of SYR in the filter blanks is likely due to background 3C\* contamination by BrC species. (b) The current procedure uses the average rate constant for SYR with the four model triplets from Kaur and Anastasio (2018). How does this average compare with the rate constant for 3DMB\*, which we used in our more recent work (e.g., Ma et al., 2023a). What are the implications for [3C\*] based on this difference? (c) The top paragraph of page 8 suggests that Kaur and Anastasio (2018) used the average of the four model triplet rate constants, but this is not correct. We used two probes to assess the average reactivity of each sample's triplets and then used a weighted rate constant specific for that reactivity. But in practice, the rate constant ended up being very close to the 3DMB\* value for most samples.*"

**Author response:** (a) The description of SYR decay in the filter blanks has been changed to "loss" instead of "photolysis" in the revised manuscript.

(b) The averaged [$^3$C\*]$_{ss}$ using the four model $^3$C\* rate constants were very close to that for the single triplet 3,4-dimethoxybenzaldehyde ($^3$DMB$^*$) as shown in Table S6. This indicated that the overall reactivity of oxidizing $^3$C\* species in our samples was also very close to the model $^3$C\*, $^3$DMB$^*$, which is consistent with the conclusion drawn in Kaur and Anastasio (2018). The following sentence has been adding in the revised manuscript:

**Page 15, line 360: "The [$^3$C\*]$_{ss}$ values were close to the values calculated using only the bimolecular rate constant for the model $^3$C\* species $^3$DMB\* (Table S6). This indicated**

that the $^3C^*$ species quantified in this study had reactivities close to $^3DMB^*$. Similar observations were reported for $^3C^*$ species in PM extracts from biomass-influenced areas in California, USA (Kaur and Anastasio, 2018; Kaur et al., 2019)."

(c) We thank the referee for pointing out our misattribution of our methodology to Kaur and Anastasio (2018). The following changes have been made to the revised manuscript:

**Page 8, line 217: "Thus, the $[^3C^*]_{ss}$ value for each extract was calculated by taking the average of the $[^3C^*]_{ss}$ values calculated using four model $^3C^*$ species (2-acetonaphthone ($^3$2AN*), 3'-methoxyacetophenone ($^3$3MAP*), 3,4-dimethoxybenzaldehyde ($^3$DMB*), and benzophenone ($^3$BP*)) which were chosen to cover the range of $^3C^*$ reactivities in atmospheric samples."**

14. Referee comment: "*Tables 1 and 2. (a) Uncertainties are 1 standard deviation? (b) The authors could simplify MAC units to $m^2$ g-$C^{-1}$ (since this is equivalent to 1E4 $cm^2$ g-$C^{-1}$). (c) Having 3 or 4 significant figures seems beyond the precision of the measurements. Is 2 sig figs a better choice? (d) Typo in Table 2 title: '-sate'.*"

**Author response:** (a) Yes. The uncertainties are one standard deviation. This information has been added in the Table footnotes. (b) We have made the requested changes in the revised manuscript. (c) We have made changes to the numbers in Tables 1 and 2. All the numbers now only have two digits after the decimal point. (d) This has been fixed in the revised manuscript.

15. Referee comment: "*Line 284. "Since the $^1O_2$ measurements were used to determine $^3C^*$ production…" It's not clear what this means.*"

**Author response:** This sentence was meant to explain the reason why the discussion of $^1O_2^*$ was placed before the discussion of $^3C^*$. We have removed this sentence from the revised manuscript to avoid any confusion.

16. Referee comment: "*Line 304. I agree that the relatively short illumination wavelengths used here (compared to simulated sunlight) are probably a major reason for the higher $^1O_2$ quantum yields, as past work has shown that photooxidant QYs tend to decrease with increasing wavelength. But on Line 345 the authors try to use the same lamp idea to also explain lower $^3C^*$ quantum yields. It seems unlikely that these two oxidants have the opposite dependence of QY on illumination wavelength. Also, in response to Line 346, Kaur et al. (2019) saw that SYR and MeJA gave similar quantum yields for oxidizing triplets, so the use of only SYR in the current work doesn't seem to be the reason for lower $^3C^*$ QYs.*"

**Author response:** We agree that it is unlikely that $^1O_2^*$ and $^3C^*$ have the opposite dependence of quantum yields on illumination wavelength and that the use of SYR as the sole $^3C^*$ probe compound is a reason for lower $^3C^*$. As such, we have made the following changes in the revised manuscript:

**Page 16, line 397: "The $\Phi_{3C^*}$ values ranged from 0.05 to 3.24 %, with a study average of (0.55 ± 0.66) % which was approximately 9 times lower than the study average of $\Phi_{1O2^*}$. The difference in $^3C^*$ and $^1O_2^*$ photosensitization efficiencies could be due to only a subset**

of $^3$C$_*$ species that can oxidize syringol being captured in our photochemical experiments since different $^3$C$_*$ species may have different photosensitization efficiencies. Our study average $\Phi_{3C*}$ was also lower than the average $\Phi_{3C*}$ (2.40 ± 1.00) %) reported by Kaur et al. (2019) for extracts of PM collected from biomass burning-influenced areas in California, USA. This suggested that the water-soluble BrC in our extracts have a lower fraction of oxidizing $^3$C$_*$ species compared to that in PM samples investigated by Kaur et al. (2019), which could be due to the different composition and age of water-soluble BrC in atmospheric PM.**

17. Referee comment: "*Line 305. What do you mean about "different methodologies"? Use of $D_2O$ versus simply using the FFA decay rate constant in water?*"

**Author response:** We meant to explain that there are different methodologies to determine $\Phi_{1O2*}$ values. While this study determined the $\Phi_{1O2*}$ values from the $R_{f,1O2*}$ and $R_{abs}$ measurements (Equation 7), other studies used a reference $^1$O$_2$$^*$ sensitizer to determine their $\Phi_{1O2*}$ values. To remove any confusion, the following changes have been made to the revised manuscript:

**Page 15, line 353: "In addition, the different methodologies used to determine $\Phi_{1O2*}$ may have contributed to our study's higher $\Phi_{1O2*}$ values. While this study determined the $\Phi_{1O2*}$ values from the $R_{f,1O2*}$ and $R_{abs}$ measurements (Equation 7), other studies used a reference $^1$O$_2$$^*$ sensitizer (e.g., perinaphthenone) to determine their $\Phi_{1O2*}$ (Manfrin et al., 2019; Bogler et al., 2022)."**

18. Referee comment: "*Lines 349 and 350. Don't these two sentences say the same thing? Lines 351 and 352. Don't these two sentences say the same thing?*"

**Author response:** The following changes have been made to the revised manuscript:

**Page 16, line 411: "The steady-state concentrations and quantum yields of $^1$O$_2$$^*$ and $^3$C$_*$ were fairly similar among the three sites (Figures S13). Variations in these values across the three sites were not statistically significant ($p > 0.05$)."**

19. Referee comment: "*Figure 3 (and all violin plots). For panel a, which of the seasonal [$^1$O$_2$] means are statistically different from each other? To what extent are any of the seasonal differences driven by differences in PM$_{2.5}$ mass concentration?*"

**Author response:** We performed student t-tests on the seasonal values for the different parameters: PM$_{2.5}$ mass/H$_2$O mass ratio, WSOC concentration, optical parameters, steady-state concentrations and quantum yields of $^1$O$_2$$^*$ and $^3$C*. Results of our t-tests have been added to the revised manuscript.

Results of our t-tests indicated that the differences in the seasonal values for the steady-state concentration of $^3$C* and the quantum yields of $^1$O$_2$$^*$ and $^3$C* were not statistically different ($p > 0.05$). These results are in line with results from the one-way ANOVA tests that we presented in the original manuscript. In contrast, we found that the winter and fall [$^1$O$_2$$^*$]$_{ss}$ values were statistically different ($p < 0.05$) from the spring and summer values. The winter [$^1$O$_2$$^*$]$_{ss}$ values

were not statistically different ($p > 0.05$) from the fall values. The spring $[^1O_2^*]_{ss}$ values were not statistically different ($p > 0.05$) from the summer values. Out of all the examined parameters: only the seasonal values for the $PM_{2.5}$ mass/$H_2O$ mass ratio and WSOC concentration matched (or were somewhat close) trends as the seasonal $[^1O_2^*]_{ss}$ values with regards to whether the difference in the parameters between two seasons was statistically significant. This suggested that the observed seasonal differences in the $[^1O_2^*]_{ss}$ values were driven primarily by the $PM_{2.5}$ mass concentration and WSOC concentration.

We have made the following changes in the revised manuscript:

**Page 17, line 437: "Overall, seasonality had noticeable effects on $[^1O_2^*]_{ss}$ and (to a lesser extent) $[^3C^*]_{ss}$, wherein these values were the highest in the fall and winter and the lowest in the summer. The seasonal trends of $[^1O_2^*]_{ss}$ and $[^3C^*]_{ss}$ correlated with the seasonal trends of the WSOC concentration and light absorption properties of water-soluble BrC (Figure S10). The fall and winter extracts had higher concentrations of and/or more absorbing water-soluble BrC comprised of organic matter of high aromaticity than the spring and summer extracts. Thus, the higher concentrations of and/or more absorbing water-soluble BrC in the winter and fall extracts likely enhanced $^1O_2^*$ and $^3C^*$ production. In particular, additional statistical analyses (Student's t-tests) performed on the seasonal values for $[^1O_2^*]_{ss}$, $PM_{2.5}$ mass/$H_2O$ mass ratio, WSOC concentration, and light absorption properties of water-soluble BrC (Table S8) suggested that the seasonal differences in the $[^1O_2^*]_{ss}$ values were driven primarily by the $PM_{2.5}$ mass concentration and WSOC concentration. Since the seasonal variations in $PM_{2.5}$ and water-soluble BrC were due to the seasonal variations in long-range air mass transport, this implied that regional $PM_{2.5}$ sources located in continental areas north of Hong Kong contributed to the higher photooxidant production in the fall and winter."**

**Table S8.** Results of t-tests performed on pairs of seasonal values for $PM_{2.5}$ mass/$H_2O$ mass ratio, WSOC concentration, light absorption properties of water-soluble BrC, and $[^1O_2^*]_{ss}$.

| $PM_{2.5}$ mass/$H_2O$ mass | Winter | Spring | Summer | Fall |
|---|---|---|---|---|
| Winter | / | Statistically significant | Statistically significant | N.S. |
| Spring | Statistically significant | / | N.S. | N.S. |
| Summer | Statistically significant | N.S. | / | Statistically significant |
| Fall | N.S. | N.S. | Statistically significant | / |

| [WSOC], $[^1O_2^*]_{ss}$ | Winter | Spring | Summer | Fall |
|---|---|---|---|---|
| Winter | / | Statistically significant | Statistically significant | N.S. |
| Spring | Statistically significant | / | N.S. | Statistically significant |
| Summer | Statistically significant | N.S. | / | Statistically significant |
| Fall | N.S. | Statistically significant | Statistically significant | / |

| $\alpha_{300}$, $R_{abs}$, $SUVA_{365}$ | Winter | Spring | Summer | Fall |
|---|---|---|---|---|
| Winter | / | Statistically significant | Statistically significant | Statistically significant |
| Spring | Statistically significant | / | N.S. | N.S. |
| Summer | Statistically significant | N.S. | / | Statistically significant |
| Fall | Statistically significant | N.S. | Statistically significant | / |

| $MAC_{300}$, $SUVA_{254}$ | Winter | Spring | Summer | Fall |
|---|---|---|---|---|

| | | | | |
|---|---|---|---|---|
| Winter | / | Statistically significant | Statistically significant | Statistically significant |
| Spring | Statistically significant | / | Statistically significant | N.S. |
| Summer | Statistically significant | Statistically significant | / | Statistically significant |
| Fall | Statistically significant | N.S. | Statistically significant | / |

**Note: The Student's t-test was used to determine whether the difference in the parameters between two seasons was statistically significant. The difference was statistically significant when $p < 0.05$. Conversely, the difference was not statistically significant (denoted as "N.S.") when $p > 0.05$. Only the parameters that were shown to be statistically significant in one-way ANOVA analysis are shown in this table. While not shown in this table, the student's t tests showed that the differences in the $[^3C^*]_{ss}$, $\Phi_{3C^*}$, and $\Phi_{1O_2^*}$ values between the different seasons were not statistically significant. Only the seasonal values for the $PM_{2.5}$ mass/$H_2O$ mass ratio and WSOC concentration matched (or had somewhat close) trends as the seasonal $[^1O_2^*]_{ss}$ values with regards to whether the difference in the parameters between two seasons was statistically significant. This suggested that the observed seasonal differences in the $[^1O_2^*]_{ss}$ values were driven primarily by the $PM_{2.5}$ mass concentration and WSOC concentration.**

20. Referee comment: "*Line 386. It's not clear what is meant by 'Even after accounting for their spread…' Line 389. '…due to their spread and standard deviations.' Isn't this saying the same thing twice? I suggest you shorten this paragraph's discussion of seasonal differences in quantum yields since there were no statistically significant differences*."

**Author response:** The "spread" was meant to indicate the one standard deviation of the average values. As requested, we have shortened the above-mentioned paragraph's discussion of seasonal differences in quantum yields:

**Page 17, line 451: "The seasonal trends of $\Phi_{1O_2^*}$ and $\Phi_{3C^*}$ (Figures 4c and 4d) were noticeably weaker than the seasonal trends of $[^1O_2^*]_{ss}$ and $[^3C^*]_{ss}$ (Figures 4a and 4b). The average $\Phi_{1O_2^*}$ for the winter, spring, summer, and fall were (5.92 ± 1.82) %, (4.07 ± 1.40) %, (4.36 ± 3.28) %, and (6.19 ± 3.22) %, respectively, while the average $\Phi_{3C^*}$ for the winter, spring, summer, and fall were (0.24 ± 1.23) %, (0.50 ± 0.40) %, (0.69 ± 0.74) %, and (0.80 ± 0.98) %, respectively. The average $\Phi_{1O_2^*}$ and $\Phi_{3C^*}$ values were noticeably the highest for the fall season. This was due to the inclusion of abnormally high quantum yield values obtained for the HT271021 sample (identified as a "far out outlier" by Tukey's fences). Fast photobleaching for the HT271021 sample during the photochemical experiments (Figures S4 and S5) likely resulted in over-estimated quantum yields. The variations in $\Phi_{1O_2^*}$ and $\Phi_{3C^*}$ across the four seasons were not statistically significant ($p > 0.05$), which indicated that seasonality did not have a significant effect on the photosensitization efficiencies of $^1O_2^*$ and $^3C^*$."**

21. Referee comment: "*Lines 506-509. This sentence is repetitious, including having the phrase '$^1O_2$ and $^3C^*$' appear three times*."

**Author response:** We have made the following changes in the revised manuscript:

**Page 23, line 587: "This necessitates the inclusion of $^3C^*$ and $^1O_2^*$ into atmospheric models since these photooxidants may play important roles in the photochemical processing of SOA in the atmospheric aqueous phases due to their high concentrations offsetting their lower reactivities."**

22. Referee comment: "*Figures S4 and S5. Column headings of the season over each column of panels (e.g., "Winter" over the first column) would help.*"

**Author response:** We have made the requested changes to the revised manuscript.

23. Referee comment: "*Figure S10. I don't see any triangles, although they're mentioned in the caption.*"

**Author response:** The triangles are supposed to indicate the outliers. However, there are no outliers in the optical datasets shown in Figure S10. Thus, we have removed the sentences referencing triangles and outliers from Figure S10's caption.

24. Referee comment: "*Table S1. The word "Set" here is confusing – doesn't it represents a single filter (sampled for 72 hr)? If so, it would be clearer to say "filter" rather than "set". As described earlier, it would be helpful to include in this Table the PM mass collected on each filter and/or the average PM$_{2.5}$ mass concentration in air over each filter period.*"

**Author response:** The word "set" in Table S1 refers to the three filters collected in one 72-h sampling period. Thus, for sample IDs that were comprised of three sets of filters (e.g., CU041220), this meant that the aggregated extracts were comprised of three consecutive 72-h sampling periods (9 days in total). For sample IDs that were comprised of two sets of filters (e.g., CU100921), this meant that the aggregated extracts were comprised of two consecutive 72-h sampling periods (6 days in total). We have added this clarification to the caption in Table S1 in the revised manuscript. As mentioned in our replies to comments 1 and 3, supplementary information about sampling methods and PM$_{2.5}$ mass concentrations have been added to Section 2.1.2 in the revised manuscript.

**References**

Bilal, M., Nichol, J. E., Nazeer, M., Shi, Y., Wang, L. C., Kumar, K. R., Ho, H. C., Mazhar, U., Bleiweiss, M. P., Qiu, Z. F., Khedher, K. M., and Lolli, S.: Characteristics of Fine Particulate Matter (PM2.5) over Urban, Suburban, and Rural Areas of Hong Kong, Atmosphere-Basel, 10, 10.3390/atmos10090496, 2019.

Bogler, S., Daellenbach, K. R., Bell, D. M., Prévôt, A. S. H., El Haddad, I., and Borduas-Dedekind, N.: Singlet Oxygen Seasonality in Aqueous PM10 is Driven by Biomass Burning and Anthropogenic Secondary Organic Aerosol, Environ Sci Technol, 56, 15389-15397, 10.1021/acs.est.2c04554, 2022.

Canonica, S. and Laubscher, H. U.: Inhibitory effect of dissolved organic matter on triplet-induced oxidation of aquatic contaminants, Photoch Photobio Sci, 7, 547-551, 10.1039/b719982a, 2008.

Chen, J., Li, C., Ristovski, Z., Milic, A., Gu, Y., Islam, M. S., Wang, S., Hao, J., Zhang, H., He, C., Guo, H., Fu, H., Miljevic, B., Morawska, L., Thai, P., Lam, Y. F., Pereira, G., Ding, A., Huang, X., and Dumka, U. C.: A review of biomass burning: Emissions and impacts on air

quality, health and climate in China, Science of The Total Environment, 579, 1000-1034, 10.1016/j.scitotenv.2016.11.025, 2017.

Chow, W. S., Liao, K. Z., Huang, X. H. H., Leung, K. F., Lau, A. K. H., and Yu, J. Z.: Measurement report: The 10-year trend of PM(2.5 )major components and source tracers from 2008 to 2017 in an urban site of Hong Kong, China, Atmospheric Chemistry and Physics, 22, 11557-11577, 10.5194/acp-22-11557-2022, 2022.

Cui, L., Wang, X. L., Ho, K. F., Gao, Y., Liu, C., Ho, S. S. H., Li, H. W., Lee, S. C., Wang, X. M., Jiang, B. Q., Huang, Y., Chow, J. C., Watson, J. G., and Chen, L. W.: Decrease of VOC emissions from vehicular emissions' in Hong Kong from 2003 to 2015: Results from a tunnel, study, Atmos Environ, 177, 64-74, 10.1016/j.atmosenv.2018.01.020, 2018.

Guo, H., Lee, S. C., Ho, K. F., Wang, X. M., and Zou, S. C.: Particle-associated polycyclic aromatic hydrocarbons in urban air of Hong Kong, Atmos Environ, 37, 5307-5317, doi.org/10.1016/j.atmosenv.2003.09.011, 2003.

Herckes, P., Valsaraj, K. T., and Collett, J. L.: A review of observations of organic matter in fogs and clouds: Origin, processing and fate, Atmospheric Research, 132-133, 434-449, https://doi.org/10.1016/j.atmosres.2013.06.005, 2013.

Herrmann, H., Schaefer, T., Tilgner, A., Styler, S. A., Weller, C., Teich, M., and Otto, T.: Tropospheric Aqueous-Phase Chemistry: Kinetics, Mechanisms, and Its Coupling to a Changing Gas Phase, Chemical Reviews, 115, 4259-4334, 10.1021/cr500447k, 2015.

Huang, X. H. H., Bian, Q. J., Ng, W. M., Louie, P. K. K., and Yu, J. Z.: Characterization of PM2.5 Major Components and Source Investigation in Suburban Hong Kong: A One Year Monitoring Study, Aerosol Air Qual Res, 14, 237-250, 10.4209/aaqr.2013.01.0020, 2014.

Kaur, R. and Anastasio, C.: First Measurements of Organic Triplet Excited States in Atmospheric Waters, Environ Sci Technol, 52, 5218-5226, 10.1021/acs.est.7b06699, 2018.

Kaur, R., Labins, J. R., Helbock, S. S., Jiang, W. Q., Bein, K. J., Zhang, Q., and Anastasio, C.: Photooxidants from brown carbon and other chromophores in illuminated particle extracts, Atmospheric Chemistry and Physics, 19, 6579-6594, 10.5194/acp-19-6579-2019, 2019.

Leresche, F., McKay, G., Kurtz, T., von Gunten, U., Canonica, S., and Rosario-Ortiz, F. L.: Effects of Ozone on the Photochemical and Photophysical Properties of Dissolved Organic Matter, Environ Sci Technol, 53, 5622-5632, 10.1021/acs.est.8b06410, 2019.

Liao, H. and Seinfeld, J. H.: Global impacts of gas-phase chemistry-aerosol interactions on direct radiative forcing by anthropogenic aerosols and ozone, Journal of Geophysical Research: Atmospheres, 110, https://doi.org/10.1029/2005JD005907, 2005.

Liao, Z., Ling, Z., Gao, M., Sun, J., Zhao, W., Ma, P., Quan, J., and Fan, S.: Tropospheric Ozone Variability Over Hong Kong Based on Recent 20 years (2000–2019) Ozonesonde Observation, Journal of Geophysical Research: Atmospheres, 126, e2020JD033054, doi.org/10.1029/2020JD033054, 2021.

Ma, L., Worland, R., Heinlein, L., Guzman, C., Jiang, W., Niedek, C., Bein, K. J., Zhang, Q., and Anastasio, C.: Seasonal variations in photooxidant formation and light absorption in

aqueous extracts of ambient particles, EGUsphere, 2023, 1-31, 10.5194/egusphere-2023-861, 2023a.

Ma, L., Worland, R., Jiang, W., Niedek, C., Guzman, C., Bein, K. J., Zhang, Q., and Anastasio, C.: Predicting photooxidant concentrations in aerosol liquid water based on laboratory extracts of ambient particles, EGUsphere, 2023, 1-36, 10.5194/egusphere-2023-566, 2023b.

Ma, L., Worland, R., Tran, T., and Anastasio, C.: Evaluation of Probes to Measure Oxidizing Organic Triplet Excited States in Aerosol Liquid Water, Environ Sci Technol, 57, 6052-6062, 10.1021/acs.est.2c09672, 2023c.

Maizel, A. C. and Remucal, C. K.: The effect of advanced secondary municipal wastewater treatment on the molecular composition of dissolved organic matter, Water Res, 122, 42-52, 10.1016/j.watres.2017.05.055, 2017.

Manfrin, A., Nizkorodov, S. A., Malecha, K. T., Getzinger, G. J., McNeill, K., and Borduas-Dedekind, N.: Reactive Oxygen Species Production from Secondary Organic Aerosols: The Importance of Singlet Oxygen, Environ Sci Technol, 53, 8553-8562, 10.1021/acs.est.9b01609, 2019.

Manfrin, A., Nizkorodov, S. A., Malecha, K. T., Getzinger, G. J., McNeill, K., and Borduas-Dedekind, N.: Reactive Oxygen Species Production from Secondary Organic Aerosols: The Importance of Singlet Oxygen, Environ Sci Technol, 53, 8553-8562, 10.1021/acs.est.9b01609, 2019.

McCabe, A. J. and Arnold, W. A.: Reactivity of Triplet Excited States of Dissolved Natural Organic Matter in Stormflow from Mixed-Use Watersheds, Environ Sci Technol, 51, 9718-9728, 10.1021/acs.est.7b01914, 2017.

Nguyen, T. K. V., Zhang, Q., Jimenez, J. L., Pike, M., and Carlton, A. G.: Liquid Water: Ubiquitous Contributor to Aerosol Mass, Environmental Science & Technology Letters, 3, 257-263, 10.1021/acs.estlett.6b00167, 2016.

Ossola, R., Jonsson, O. M., Moor, K., and McNeill, K.: Singlet Oxygen Quantum Yields in Environmental Waters, Chemical Reviews, 121, 4100-4146, 10.1021/acs.chemrev.0c00781, 2021.

Yang, J., Ma, L., He, X., Au, W. C., Miao, Y., Wang, W. X., and Nah, T.: Measurement report: Abundance and fractional solubilities of aerosol metals in urban Hong Kong – insights into factors that control aerosol metal dissolution in an urban site in South China, Atmos. Chem. Phys., 23, 1403-1419, 10.5194/acp-23-1403-2023, 2023.

Zhang, X. X., Yuan, Z. B., Li, W. S., Lau, A. K. H., Yu, J. Z., Fung, J. C. H., Zheng, J. Y., and Yu, A. L. C.: Eighteen-year trends of local and non-local impacts to ambient PM10 in Hong Kong based on chemical speciation and source apportionment, Atmospheric Research, 214, 1-9, 10.1016/j.atmosres.2018.07.004, 2018.